# On Adaptive Attacks
# to Adversarial Example Defenses

**Florian Tramèr**[*]
Stanford University
tramer@cs.stanford.edu

**Nicholas Carlini**[*]
Google
nicholas@carlini.com

**Wieland Brendel**[*]
University of Tübingen
wieland.brendel@uni-tuebingen.de

**Aleksander Mądry**
MIT
madry@mit.edu

## Abstract

Adaptive attacks have (rightfully) become the de facto standard for evaluating defenses to adversarial examples. We find, however, that typical adaptive evaluations are incomplete. We demonstrate that thirteen defenses recently published at ICLR, ICML and NeurIPS—and which illustrate a diverse set of defense strategies—can be circumvented *despite* attempting to perform evaluations using adaptive attacks. While prior evaluation papers focused mainly on the end result—showing that a defense was ineffective—this paper focuses on laying out the methodology and the approach necessary to perform an adaptive attack. Some of our attack strategies are generalizable, but no single strategy would have been sufficient for all defenses. This underlines our key message that adaptive attacks *cannot* be automated and always require careful and appropriate tuning to a given defense. We hope that these analyses will serve as guidance on how to properly perform adaptive attacks against defenses to adversarial examples, and thus will allow the community to make further progress in building more robust models.

## 1 Introduction

Over the last five years the research community has attempted to develop defenses to adversarial examples [SZS+14, BCM+13]. This has proven extraordinarily difficult. Indeed, a common theme has been proposing defenses that—due to having been tested only against static and relatively weak attacks—were promptly circumvented by a stronger attack [CW17a, ACW18].

Recent community efforts and guidelines to improve defense evaluations have had a noticeably positive impact. In particular, there has been a significant uptake of evaluations against *adaptive attacks*, i.e., attacks that were specifically designed to target a given defense—the ratio of defenses evaluated against adaptive attacks has increased from close to zero in 2017 [CW17a] to one third in 2018 [ACW18] and to nearly all of them today.[1] This leads to the question:

*With their much-improved evaluation practices, are these defenses truly robust?*

We find that this is *not* the case. Specifically, in an analysis of thirteen defenses, selected from recent ICLR, ICML, and NeurIPS conferences to illustrate diverse defensive strategies, we find that we can circumvent *all* of them and substantially reduce the accuracy from what was originally claimed.

---

[*]Equal contribution

[1]There is a similarly positive trend in terms of releasing source code for defenses. In particular, *every* defense we analyzed either released source code, or made it available upon request.

Importantly, while almost all of these defenses performed an evaluation involving adaptive attacks, these evaluations ended up not being sufficient. For example, it is common for papers to repurpose existing "adaptive" attacks that circumvented some prior defense, without considering how to change it to target the new defense. We suspect that this shortcoming might have been caused, in part, by the fact that prior work on circumventing defenses typically shows only the final, successful attack, without describing the methodology that was used to come up with this attack and thus leaving open questions such as "How was the attack discovered?" or "What other attacks were unsuccessful?".

To remedy this problem, instead of merely demonstrating that the thirteen defenses we studied can be circumvented by stronger attacks, we actually walk the reader through our full process of analyzing each defense, from an initial paper read-through, to our hypotheses about what would be required for the defense to be circumvented, to an ultimately successful attack. This approach lets us more clearly document the many steps involved in developing a strong adaptive attack.

The goal of our analyzes is not to reduce analyzed model's accuracy all the way to $0\%$.[2] Instead, we want to demonstrate that the existing adaptive attack evaluation methodology has shortcomings, and that stronger adaptive attacks can (at least partially) degrade each defense's accuracy from what is reported in the original evaluations.

Whereas prior work often needed to develop new techniques [ACW18] to evade defenses, we find that the technical tools to evaluate defenses properly already exist. A better attack can be built using only tools that are well-known in the literature. Thus, the issue with current defense evaluations is methodological rather than technical.

After describing our methodology (Section 3) and providing an overview of common themes distilled from our evaluations (Section 4), we give a summary of the evaluation of each defense (Section 5), leaving the full evaluation for the appendix due to space constraints. We state and test our initial hypotheses—gathered from reading the paper and source code—as to why the original evaluation may have been insufficient. Finally, we describe how the observations made throughout our evaluation inform the design of a final adaptive attack that succeeds in circumventing the defense.

An overarching theme of our evaluations is simplicity. Attacks need not be complicated, even when the defense is. There are often only a small number of important components in complex defenses—carefully targeting these can lead to simpler and stronger attacks. We design our loss functions, the cornerstone of successful adaptive attacks, so that they are easy to optimize and *consistent*—so that higher loss values result in strictly stronger attacks. While some of our techniques are generalizable, no single attack strategy would have sufficed for all defenses. This underlines the crucial fact that adaptive attacks *cannot* be automated and always require appropriate tuning to a each defense.

In the spirit of responsible disclosure, we contacted all the authors of the defenses we evaluated prior to this submission, and offered to share code or adversarial examples to enable independent verification of our claims. In all but one case (where the defense authors did not respond to our original message), the authors acknowledged our attacks' effectiveness. To promote reproducibility and encourage others to perform independent re-evaluations of proposed defenses, we release code for all of our attacks at `https://github.com/wielandbrendel/adaptive_attacks_paper`.

## 2 Background

This paper studies the robustness of defenses to adversarial examples. Readers familiar with the relevant literature and notation [SZS+14, CW17b, MMS+17, ACW18] can continue with Section 3 where we describe our methodology.

For a classification neural network $f$ and a natural input $x$ (e.g., drawn from the test set) with a true label $y$, an *adversarial example* [SZS+14] is a perturbed input $x'$ such that: (1) $\|x' - x\|$ is small, for some distance function[3] $\|\cdot\|$ but (2) it is classified incorrectly either *untargeted*, so that $f(x') \neq y$, or *targeted*, so that $f(x') = t$ for some selected class $t \neq y$.[4]

To generate an adversarial example, we construct a *loss function* $L$ so that $L(x, y)$ is large when $f(x) \neq y$, and then maximize $L(x', y)$ while keeping the perturbation $\|x - x'\|$ small. Defining an appropriate loss function $L$ is the key component of any adaptive attack. The typical starting-point for any loss function is the cross-entropy loss $L_{CE}$.

While there are many methods for generating adversarial examples, the most widely adopted approach is gradient descent on input space [CW17b, MMS$^+$17]. Let $x_0 = x$ and then repeatedly set

$$x_{i+1} = \texttt{Proj}(x_i + \alpha \cdot \texttt{normalize}(\nabla_{x_i} L(x_i, y)))$$

where `Proj` projects the inputs onto a smaller domain to keep the distortion small, `normalize` enforces a unit-length step size under the considered norm, and $\alpha$ controls the size of steps that are taken. After $N$ steps (e.g., 1000), we let $x' = x_N$ be the resulting adversarial example. Further background on common adversarial examples generation techniques is provided in Appendix A.

In some cases, it will be useful to represent the classifier in the form $f(x) = \arg\max z(x)$, where $z(x)$ is a vector of class-scores which we refer to as the logit layer (e.g., before any softmax activation function); thus, $z(x)_i$ is the logit value for class $i$ on input $x$. Through a slight abuse of notation, we sometimes use $f(x)$ to refer to either the classifier's output class (i.e., $f(x) = y$) or to the full vector of class probabilities (i.e., $f(x) = [p_1, \ldots, p_K]$). The choice should be clear from context.

All defenses we study in this paper claim *white-box* robustness: here, the adversary is assumed to have knowledge of the model architecture, weights, and any necessary additional information. This allows us to use the above attack techniques.

## 3  Methodology

The core of this paper documents the attacks we develop on the thirteen defenses. Due to space constraints, our evaluations are summarized in Section 5 and provided in full in the appendix. Our structure for each evaluation section is as follows:

- **How the defense works.** We begin with a brief description of the defense and introduce the necessary components. We encourage readers to pause after this section to reflect on how one might attack the described defense before reading our method.[5]

- **Why we selected this defense.** While we study many defenses in this paper, we cannot hope to evaluate *all* defenses published in the last few years. We selected each defense with two criteria: (1) it had been accepted at ICLR, ICML, or NeurIPS; indicating that independent reviewers considered it interesting and sound, and (2) it clearly illustrates some concept; if two defenses were built on a same idea or we believed would fail in identical situations we selected only one.[6]

- **Initial hypotheses and experiments.** We fist read each paper with a focus on finding potential reasons why the defense might still be vulnerable to adversarial examples, in spite of the robustness evaluation initially performed in the paper. To develop additional attack hypotheses, we also studied the defense's source code, which was available publicly or upon request for *all* defenses.

- **Final robustness evaluation.** Given our candidate hypotheses of why the defense could fail, we turn to developing our own adaptive attacks. In most cases, this consists of (1) constructing an improved loss function where gradient descent could succeed at generating adversarial examples, (2) choosing a method of minimizing that loss function, and (3) repeating these steps based on new insights gleaned from an attack attempt.

- **Lessons learned.** Having circumvented the defense, we look back and ask what we have learned from this attack and how it may apply in the future.

---

perform untargeted attacks (because a model robust to untargeted attacks is also robust to targeted attacks). We intend for this paper to be a study in how to perform adaptive attacks evaluations; as such, we choose the untargeted attack objective that future defense creators will typically choose.

[5]To facilitate the reading of this paper, we use the standard notation defined in Section 2 to describe each defense and attack. Each section introduces additional defense-specific notation only if needed. Our notation thus often differs slightly from that used in each defense's original paper.

[6]We discovered *a posteriori* that some of the selected defenses can fail in similar ways. This report contains our evaluations for *all* of our initially selected defenses. In particular, we did not cherry-pick the defense evaluations to present based on the results of our evaluation.

|  | Defense | Attack Themes | | | | | |
|---|---|---|---|---|---|---|---|
|  |  | T1 | T2 | T3 | T4 | T5 | T6 |
| Appendix B | *k-Winners Take All* [XZZ20] | ● |  |  |  | ● | ● |
| Appendix C | *The Odds are Odd* [RKH19] |  |  | ● | ● |  |  |
| Appendix D | *Generative Classifiers* [LBS19] |  | ● | ● |  |  |  |
| Appendix E | *Sparse Fourier Transform* [BMV18] | ● | ● |  |  |  |  |
| Appendix F | *Rethinking Cross Entropy* [PXD+20] |  |  | ● |  | ● |  |
| Appendix G | *Error Correcting Codes* [VS19] | ● | ● |  |  |  |  |
| Appendix H | *Ensemble Diversity* [PXD+19] |  |  |  |  | ● |  |
| Appendix I | *EMPIR* [SRR20] |  | ● |  |  | ● |  |
| Appendix J | *Temporal Dependency* [YLCS19] | ● |  | ● | ● | ● |  |
| Appendix K | *Mixup Inference* [PXZ20] | ● |  |  |  |  |  |
| Appendix L | *ME-Net* [YZKX19] | ● | ● |  | ● |  | ● |
| Appendix M | *Asymmetrical Adv. Training* [YKR20] |  |  | ● | ● |  | ● |
| Appendix N | *Weakness into a Strength* [HYG+19] | ● | ● | ● | ● |  |  |

Table 1: The attack themes illustrated by each defense we evaluate.

The flow of each section directly mirrors the steps we actually took to evaluate the robustness of the defense, without retroactively drawing conclusions on how we wish we had discovered the flaws. We hope that this will allow our evaluation strategy to serve as a case study for how to perform future evaluations from start to finish, rather than just observing the end result.

We emphasize that our main goal is to assess whether each defense's initial robustness evaluation was appropriate, and—if we find it was not—to demonstrate how to build a stronger adaptive attack on that defense. Importantly, we do not wish to claim that the robustness techniques used by many of these defenses hold no merit whatsoever. It may well be that some instantiations of these techniques can improve model robustness. To convincingly make such a claim however, a defense has to be backed up by a strong adaptive evaluation—the focal point of our tutorial.

## 4 Recurring Attack Themes

We identify six themes (and one meta-theme) that are common to multiple evaluations. We discuss each below. Table 1 gives an overview of the themes that relate to each of the defenses we studied.

**(meta-theme) T0: Strive for simplicity.** Our adaptive attacks are consistently *simpler* than the adaptive attacks evaluated in each paper. In all cases, our attack is as close as possible to straightforward gradient descent with an appropriate loss function; we add additional complexity only when simpler attacks fail. It is easier to diagnose failures of simpler attacks, in order to iterate towards a stronger attack. The themes below mainly capture various ways to make an attack simpler, yet stronger.

**T1: Attack (a function close to) the full defense.** If a defense is end-to-end differentiable, the entire defense should be attacked. Additional loss terms should be avoided unless they are necessary; conversely, any components, especially pre-processing functions, should be included if at all possible.

**T2: Identify and target important defense parts.** Some defenses combine many sub-components in a complex fashion. Often, inspecting these components reveals that only one or two would be sufficient for the defense to fail. Targeting only these components can lead to attacks that are simpler, stronger, and easier to optimize.

**T3: Adapt the objective to simplify the attack.** There are many objective functions that can be optimized to generate adversarial examples. Choosing the "best" one is not trivial but can vastly boost an attack's success rate. For example, we find that targeted attacks (or "multi-targeted attacks" [GUQ+19]) are sometimes easier to formulate than untargeted attacks. We also show that *feature adversaries* [SCFF16] are an effective way to circumvent many defenses that *detect* adversarial examples. Here, we pick a natural input $x^*$ from another class than $x$ and generate an adversarial example $x'$ that matches the feature representation of $x^*$ at some inner layer.

**T4: Ensure the loss function is consistent.** Prior work has shown that many defense evaluations fail because of loss functions that are hard to optimize [CW17b, ACW18]. We find that many defense evaluations suffer from an orthogonal and deeper issue: the chosen loss function is not actually a good proxy for attack success. That is, even with an all-powerful optimizer that is guaranteed to find the loss' optimum, an attack may fail. Such loss functions are not *consistent* and should be avoided.

**T5: Optimize the loss function with different methods.** Given a useful loss function that is as simple as possible, one should choose the appropriate attack algorithm as well as the right hyper-parameters (e.g. sufficiently many iterations or repetitions). In particular, a white-box gradient-based attack is not always the most appropriate algorithm for optimizing the loss function. In some cases, we find that applying a "black-box" score-based attack [CZS$^+$17, IEAL18] or decision-based attack [BRB18] can optimize loss functions that are difficult to differentiate.

**T6: Use a strong adaptive attack for adversarial training.** Many papers claim that combining their defense with adversarial training [SZS$^+$14, MMS$^+$17] yields a stronger defense than either technique on its own. Yet, we find that many such combined defenses result in *lower* robustness than adversarial training alone. This reveals another failure mode of weak adaptive attacks: if the attack used for training fails to reliably find adversarial examples, the model will not resist stronger attacks.

# 5 Our Adaptive Evaluations

We evaluate and circumvent thirteen defenses. In this section we describe, briefly, how each defense works and what we need to do in order to circumvent it. The ordering of defenses is arbitrary. Due to space constraints, the full evaluation for each defense is contained in the appendix.

## 5.1 k-Winners Take All

This defense [XZZ20] intentionally masks gradients by replacing standard ReLU activations with a discontinuous k-Winners-Take-All ($k$-WTA) function, which sets all but the $k$ largest activations of a layer to zero. As a result, even small changes to the input can drastically change the activation patterns in the network and lead to large jumps in predictions. As the model is not locally smooth, it is hard to find adversarial examples with standard gradient-based attacks.

Whenever a defense gives rise to such semi-stochastic discontinuous jumps in model predictions, our first hypothesis is that we need to find a smooth approximation of the model's decision surface that allows for gradient descent to succeed. Even though this defense is deterministic, we adopt a strategy similar to the EOT technique [AEIK18] (see Appendix A) that is commonly used to smooth out variations in randomized defenses. We estimate smoothed gradients via finite-differences from 20,000 small perturbations of the input, and run PGD for 100 steps. This reduces accuracy from 50.0% to 0.16% for an adversarially trained CIFAR-10 model with k-WTA activations. Hence, adding k-WTA activations actually *decreases* robustness compared to the adversarial training baseline.

## 5.2 The Odds are Odd

This defense [RKH19] detects adversarial examples based on the distribution of a classifier's logit values. On input $x$, the defense compares the logit vector $z(x)$ to a "noisy" logit vector $z(x + \delta)$, where $\delta \sim \mathcal{N}$ is sampled at test-time from some distribution $\mathcal{N}$ (e.g., a multinomial Gaussian). The assumption is that for clean examples, $z(x) \approx z(x + \delta)$, while for adversarial examples the two logit vectors will differ significantly. The defense in Section 5.13 has a similar rationale.

The authors evaluated standard attacks combined with EOT [AEIK18] to average over the defense's randomness. Yet, by design, many attacks find adversarial examples that are either not robust to noise (e.g., C&W [CW17b]) or *too* robust to noise (e.g., PGD [MMS$^+$17]), compared to clean inputs. To create adversarial examples with similar noise robustness as clean inputs, we use "feature adversaries" [SCFF16]: we pick a clean input $x^*$ from another class than $x$, and minimize $||z(x') - z(x^*)||_2^2$ so as to match the logits of the clean input. This attack reduces the defense's accuracy to 17% on CIFAR-10 ($\epsilon = 8/255$). Further including EOT reduces accuracy to 0%.

## 5.3 Are Generative Classifier More Robust?

This defense [LBS19] is based on the Variational Auto-Encoder framework [KW14]. It is fairly complex and displays many characteristics of what makes an evaluation challenging, such as the use of multiple models, aggregation of multiple losses, stochasticity, and an extra detection step.

The defense first runs the input through a set of stochastic encoder-decoder models. It then computes four loss terms per class, and combines these to form a likelihood score for each class. Finally, the defense rejects inputs with "unusual" class scores, according to a KL-divergence test.

The authors' evaluate attacks that optimize over all the defense's components, and which fail to find adversarial examples even for large perturbations. By inspecting the model's four loss terms, we found that only one term has a strong influence on the class scores. Optimizing only that loss yields a simpler attack with a high success rate. To evade the detection step, we use "feature adversaries" [SCFF16] and create adversarial examples whose class scores match those of clean examples from other classes. This attack reduces the defense's accuracy to $1\%$ on CIFAR-10 ($\epsilon = 8/255$).

### 5.4 Robust Sparse Fourier Transform

This defense [BMV18] prevents $\ell_0$ adversarial examples by "compressing" each image and projecting to the top-$k$ coefficients of the discrete cosine transform. This paper does not contain an analysis of its robustness to adaptive attacks. As such, we find that implementing the defense in a straightforward differentiable manner and then generating $\ell_0$ adversarial examples (using [CW17b]) on this combined function succeeds. Our attack generates adversarial examples on the defended classifier with a median distortion of 14.8 pixels, comparable to the 15 pixels reported by [CW17b] for an undefended model.

### 5.5 Rethinking Softmax Cross Entropy

This defense [PXD$^+$20] uses a new loss function during training in order to increase adversarial robustness. The authors propose the *Max-Mahalanobis center (MMC) loss*, an $\ell_2$ loss where there is a target vector $\mu_y$ for each class label, and the classification is (roughly) given by $\arg\min_y (f(x) - \mu_y)^2$.

In general, whenever a defense changes the loss function used to train a model, our initial hypothesis is that the new model has a loss surface that is not well suited to attacks with the standard softmax cross-entropy loss. We find that this is the case here. Instead of indirectly optimizing the softmax cross-entropy loss, we directly optimize the $\ell_2$ loss between $f(x)$ and the target vectors $\mu_y$. This attack reduces the classifier accuracy to under $0.5\%$ at distortion $\epsilon = 0.031$ on CIFAR-10.

### 5.6 Error Correcting Codes

This defense [VS19] proposes a method for training an ensemble of models with sufficient diversity that the redundancy can act as error correcting codes to allow for robust classification. Instead of each model in the ensemble having the same output label distribution, all models are binary classifiers on a random partitioning of the classes. The output of the ensemble is the majority vote of the ensemble.

This paper does not contain an adaptive evaluation. This shows in the strong claims: the paper reports nearly $40\%$ accuracy at an $\ell_\infty$ distortion of $\varepsilon = 0.5$—a tell-tale sign of gradient masking [ACW18]. To attack the defense, we first remove some components that make gradient signals brittle (e.g., successive sigmoid and log operations). By running a multi-targeted attack [GUQ$^+$19], we reduce the ensemble accuracy to $5\%$ at $\varepsilon = 0.031$, compared to the reported $57\%$ accuracy at this distortion.

### 5.7 Ensemble Diversity

This defense [PXD$^+$19] trains an ensemble of models with a regularization term that encourages diversity among the models' predictions. Ensembles of diverse models are posited to be more robust than single models or less diverse ensembles. The rationale is similar to the defense in Section 5.6.

As the defense relies on a change of training objective that does not appear to cause gradient masking, no adaptive changes to standard gradient descent attacks seems necessary. The authors evaluated a wide range of attacks such as PGD [MMS$^+$17], BIM [KGB16] and MIM [DLP$^+$18], but these attacks are all very similar. Moreover, the attacks were run for only 10 steps with a step size of $\epsilon/10$, and thus likely failed to converge. We found that multiplying the step size for PGD by three reduces accuracy from $48\%$ to $26\%$. Increasing the number of steps to 250 further reduces accuracy to $10\%$. Repeating the attack three times on each sample reduces accuracy to $7\%$. Finally, for the remaining samples we applied the B&B attack [BRM$^+$19] (which was not available when this defense was published) with 20 steps and five repetitions, to reduce accuracy to $0\%$ (on CIFAR-10 with $\epsilon = 0.01$).

## 5.8 EMPIR

This defense [SRR20] is the third one we study that develops a method to defend against adversarial examples by constructing an ensemble of models, but this time with mixed precision of weights and activations. This paper trains three models $\{f_i\}$ with different levels of precision on the weights and on the activations for each, and returns the output as the majority vote of the models.

This paper does not contain an explicit analysis of its robustness to adaptive attacks, but performs an analysis of standard gradient-based attacks against the ensemble. Investigating the authors' code, we found that their attack *does* include a BPDA-style estimate of the gradient. Our initial hypothesis was that performing gradient descent with respect to the entire model ensemble and ensuring that gradient information was correctly propagated through the model would suffice to bypass this defense.

Because taking a majority vote can be numerically unstable, we take the probability vectors of the three models $f_1(x), f_2(x), f_3(x)$ and average them component-wise to obtain a model $\hat{f}$ whose prediction for class $i$ is given by $\hat{f}(x)_i = (f_1(x)_i + f_2(x)_i + f_3(x)_i)/3$. We then run PGD on the cross-entropy loss for $\hat{f}$. With 100 iterations of PGD, we reduce the defense's accuracy to 1.5% at $\varepsilon = 0.031$ (the original evaluation reported accuracy of 13.5% at $\varepsilon = 0.1$ with 40 iterations of PGD).

## 5.9 Temporal Dependency

This defense [YLCS19] detects adversarial examples for speech recognition. It checks if the classification of the first half of an audio sample is similar to the first half of the classification of the full audio sample. That is, if $x_{\text{pre}}$ is a prefix of the sequence $x$, then this defense checks that $f(x)_{\text{pre}} \approx f(x_{\text{pre}})$.

The paper performs an extensive adaptive evaluation, trying three different attack techniques. One of these attacks was similar to our own initial hypothesis for a successful attack: select a target sequence $t$, and minimize the transcription loss on both the prefixed input $f(x_{\text{pre}})$ and on the full input $f(x)$, where the loss on the prefixed input is computed between the classification of $f(x_{\text{pre}})$ and the prefix of the target sequence $t$. However, we were confident that there was *some* error in the original evaluation because the success rate of the attack is low for an *unbounded* attack: even without placing a bound on the adversarial distortion, the attack does not succeed 100% of the time at fooling the detector.

We found a number of details that, when combined, dramatically increased the attack success rate to 100% with small distortion. The most important change we made is in the way that the prefix function is calculated. Specifically, we optimize the loss function $L(x') = L(f(x'), t) + L(f(x'_{\text{pre}}), t_{\text{pre}})$, where $t_{\text{pre}}$ is the first $l$ characters of the *true* prediction $t$ where $l = \text{len}(f(x'_{\text{pre}}))$. Importantly, this is *not* just the first $l$ characters of $f(x')$: this value might not be the correct output at any point in time.

On a baseline classifier, we require a median $\ell_\infty$ distortion of 41 (out of a 16-bit integer range of 65536). For the classifier defended with this defense, we require a distortion of 46, a slight increase but not significantly higher than baseline defenses (e.g., bit quantization).

## 5.10 Mixup Inference

This defense [PXZ20] mitigates the effect of adversarial perturbations via a random interpolation procedure at inference. On input $x$, the defense computes $K$ linear interpolations $\hat{x}_k = \lambda x + (1-\lambda)s_k$ with other random samples $s_k$, and averages the model's predictions on all $K$ interpolations.

The authors' original evaluation does consider an adaptive attack that, however, performs no better than other non-adaptive attacks. The considered adaptive attack averages the adversarial examples generated by multiple non-adaptive attacks, and thus does not directly account for the defense's special inference procedure. We propose an adaptive attack that combines PGD with an EOT [AEIK18] procedure to account for the defense's randomness: in each PGD step, we estimate the gradient of the full defense by averaging gradients obtained from multiple randomly interpolated samples. Our attack reduces accuracy of the strongest defense model (which is combined with adversarial training) from 57% to 43.9% on CIFAR-10, which is close to the robustness of the baseline model (42.5%).

## 5.11 ME-Net

This defense [YZKX19] uses a pre-processing step that randomly discards pixels in an image, and then reconstructs the image using matrix-estimation. The defense trains a model on such pre-processed inputs, with the aim of learning representations that are less sensitive to small input variations.

The authors' evaluation uses BPDA [ACW18] to approximate gradients through the matrix-estimation step. We show that the attack can be strengthened by also using EOT (see Appendix A), to account for the random choice of discarded pixels. Using both BPDA and EOT reduces the defense's accuracy to 15% on CIFAR-10 ($\epsilon = 8/255$). The attack could not be improved further *even when we removed the pre-processing step entirely*, thereby suggesting that the learned model is indeed somewhat robust.

The authors also combined their defense with adversarial training. We find that this model is *less* robust than one trained solely with adversarial training—presumably because the original BPDA attack (without EOT) also fails to find strong adversarial examples on the model at training time.

## 5.12 Asymmetrical Adversarial Training

This defense [YKR20] uses adversarially-trained models to detect adversarial examples. In its "integrated classifier" mode, the defense first uses a base classifier $f$ to output a prediction $f(x) = y$. The input $x$ is then fed to a detector for that class, $h_y$, and rejected if $h_y(x)$ is below some threshold. Each detector $h_i$ is adversarially trained to reject perturbed inputs from classes other than $i$.

The defense is very simple and thus unlikely to suffer from gradient masking. The authors propose a natural adaptive attack, that aims to jointly fool the base classifier $f$ and all the detectors $h_i$. The issue is that this attack "over-optimizes" by trying to fool all detectors instead of just the one for the predicted class. We show that formulating a "multi-targeted" attack [GUQ+19] is very simple and more effective: for every target class $t \neq y$, we jointly maximize $f(x)_t$ (to get a misclassification as $t$) and $h_t(x)$ (to fool the detector for class $t$). We then pick the best target class for each input.

We reduce the defense's accuracy from 30% to 11% on CIFAR-10 ($\epsilon = 8/255$). The authors also propose a defense that simply combines the robust detectors and outputs $y = \arg\max_i h_i(x)$. We reduce this defense's accuracy from 55% to 37%, slightly below (standard) adversarial training.

## 5.13 Turning a Weakness into a Strength

This defense [HYG+19] detects that an input is adversarial via two tests: (1) perturbing the input with noise does not result in classification changes, and (2) the input's nearest adversarial example is farther away than some threshold. This defense is conceptually very similar to the one in Section 5.2.

While the defense is conceptually simple, the original evaluation methodology is complex and constructing a single unified loss function that encodes the attacker's objective is challenging. In particular, the authors develop a 4-way weighted loss with two components that are non-differentiable and so the paper applies BPDA [ACW18]. We believe that BPDA is not a method that should be applied as a first attempt whenever a non-differentiable loss is encountered, but rather as a last resort.

The defense relies on the assumption that there are no inputs that (a) have high confidence on random noise, but (b) are still close to the decision boundary when running an adversarial attack. We explicitly construct inputs with these two properties, by first generating an adversarial example $x'$ with high confidence, and then performing a binary search between $x'$ and the input $x$. This attack reduces the defense's accuracy to $< 1\%$ at a $0\%$ detection rate on both CIFAR-10 and ImageNet, for the same threat models as in the original paper.

## 6    Conclusion

We see a marked improvement in defense evaluations compared to those studied in prior work [CW17a, ACW18]: whereas in the past, papers would simply not perform an evaluation with a strong adaptive attack, nearly all defense evaluations we study *do* perform adaptive attacks that aim to evade the proposed defense. However, we find that despite the *presence* of an adaptive attack evaluation, all thirteen defenses we analyzed could be circumvented with improved attacks.

We have described a number of attack strategies that proved useful in circumventing different defenses. Yet, we urge the community to refrain from using these adaptive attacks as templates to be copied in future evaluations. To illustrate why, we propose the following informal "no-free-lunch-theorem":

*For any proposed attack, it is possible to build a non-robust defense that prevents that attack.*

Such a defense is generally not interesting. We thus encourage viewing robustness evaluations against prior attacks (including adaptive attacks on prior defenses) only as a useful sanity check. The main focus of a robustness evaluation should be on developing comprehensive adaptive attacks that explicitly uncover and target the defense's weakest links (by making use of prior techniques only if appropriate). In particular, we strongly encourage authors to focus their evaluation **solely** on adaptive attacks (and defer evaluations based on non-adaptive attacks to an appendix). It is nevertheless important to include a non-adaptive evaluation to demonstrate that simple methods are insufficient. We hope that this paper can serve as a guide to performing and evaluating such adaptive attacks.

It is critical that adaptive attacks are hand-designed to target specific defenses. No automated tool will be able to comprehensively evaluate the robustness of a defense, no matter how sophisticated. For example, the state-of-the-art in automated attacks, "AutoAttack" [**?** ], evaluates four of the same defenses we do. In two cases, AutoAttack only partially succeeds in attacking the proposed defenses when we break them completely. Furthermore, existing automated tools like AutoAttack cannot be directly applied to many of the defenses we studied, such as those that detect adversarial examples.

Perhaps the most important lesson from our analysis is that adaptive attacks should be as *simple* as possible, while resolving any potential optimization difficulties. Each new component added on top of straightforward gradient descent is another place where errors can arise. On the whole, the progress in defense evaluations over the past years is encouraging, and we hope that stronger adaptive attacks evaluations will pave the way to more robust models.

## Broader Impact

Research that studies the security of machine learning, and especially papers whose primary purpose is to develop attacks on proposed defenses, must be careful to do more good than harm. We believe that this will be the case with our paper. After decades of debate, the computer security community has largely converged on "responsible disclosure" as the optimal method for disclosing vulnerabilities: after discovering a vulnerability, responsible disclosure dictates that the affected parties should be notified fist, and after a reasonable amount of time, the disclosure should be made public so that the community as a whole can learn from it.

We notified all authors of our breaks of their defenses before making our paper public. Authors from twelve of the thirteen papers responded to us and verified that our evaluations were accurate (we offered to provide the generated adversarial examples to all authors). Further, we do not believe that there are any deployed systems that rely on the security of any of these particular defenses.

However, it remains a possibility that our methodology for constructing attacks could be used to break some other system which has been deployed, or will be deployed in the future. This is unavoidable, however we firmly believe that the help that our paper can provide to researchers designing new defenses significantly outweighs the help that it may provide an actual malicious actor. Our paper is focused on assisting researchers perform more thorough evaluations, and diagnosing failures in evaluations—not on attacking real systems or users.

## Acknowledgments and Disclosure of Funding

We thank the authors of all the defenses we studied for helpful discussions, for sharing code and pre-trained models, and for comments on early drafts of this paper, and the anonymous reviewers for their comments.

Florian Tramèr's research was supported in part by the Swiss National Science Foundation (SNSF project P1SKP2_178149). Nicholas Carlini's research as supported by Google. Wieland Brendel's research was supported by the German Federal Ministry of Education and Research through the Tübingen AI Center (FKZ 01IS18039A) as well as by the Intelligence Advanced Research Projects Activity (IARPA) via Department of Interior/Interior Business Center (DoI/IBC) contract number

D16PC00003. The U.S. Government is authorized to reproduce and distribute reprints for Governmental purposes notwithstanding any copyright annotation thereon. Disclaimer: The views and conclusions contained herein are those of the authors and should not be interpreted as necessarily representing the official policies or endorsements, either expressed or implied, of IARPA, DoI/IBC, or the U.S. Government. Aleksander Mądry's research was supported in part by the NSF awards CCF-1553428 and CNS-1815221. This material is based upon work supported by the Defense Advanced Research Projects Agency (DARPA) under Contract No. HR001120C0015.

## Footnotes

[2]That said, most of the defenses we study are ineffective at increasing robustness even partially, and we are able to reduce the accuracy of the defended classifier to the accuracy of a baseline model.

[3]We evaluate each defense under the threat model for which robustness was claimed; in all cases $\ell_p$ norms for various $p \geq 0$.

[4]Most papers evaluate on whichever result is stronger: thus, papers that propose attacks perform targeted attacks (because if a targeted attack succeeds, then so will untargeted attacks) and papers that propose defenses

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
