[Supplementary Material · paper_supplement_7887.pdf]

# A   Background on Procedures for Generating Adversarial Examples

This section provides some additional background on common attack strategies used in defense evaluations.

**PGD.**   Projected Gradient Descent [MMS$^+$17] is a strategy for finding an adversarial example $x'$ for an input $x$ that satisfies a given norm-bound $\|x' - x\|_p \leq \epsilon$.

Let $B$ denote the $\ell_p$-ball of radius $\epsilon$ centered at $x$. The attack starts at a random point $x_0 \in B$, and repeatedly sets

$$x_{i+1} = \texttt{Proj}_B(x_i + \alpha \cdot g)$$
$$\text{for } g = \arg\max_{\|v\|_p \leq 1} v^\top \nabla_{x_i} L(x_i, y) \ .$$

Here, $L(x, y)$ is a suitable loss-function (e.g., cross-entropy), $\alpha$ is a step-size, $\texttt{Proj}_B$ projects an input onto the norm-ball $B$, and $g$ is the *steepest ascent* direction for a given $\ell_p$-norm. E.g., for the $\ell_\infty$-norm, $\texttt{Proj}(z)$ is a clipping operator and $g = \texttt{sign}(\nabla_{x_i} L(x_i, y))$ .

**C&W.**   [CW17b] propose a number of attacks commonly referred to as C&W attacks. Instead of maximizing a loss function $L(x, y)$ under a given perturbation constraint (as is done with PGD above), these attacks aim to find the *smallest* successful adversarial perturbation. That is, the attack maximizes the following objective:

$$\text{maximize}_{x'} \ L(x', y) - \lambda \cdot \|x' - x\|_p \ .$$

The parameter $\lambda > 0$ balances the objectives of maximizing the loss and minimizing the perturbation size. A binary-search over $\lambda$ is used to find the minimally perturbed $x'$ that results in a successful attack.

**BPDA.**   The Backward Pass Differentiable Approximation (BPDA) is a gradient approximation technique introduced by [ACW18]. It is suited for defenses that have one or more non-differentiable components.

Let $f$ be a $n$-layer network $f(x) = f^n \circ f^{n-1} \circ \cdots \circ f^1(x)$ where layer $f^i(\cdot)$ is non-differentiable (or hard to differentiate). To approximate the gradient $\nabla_x f(x)$, we first find a differentiable function $g(x)$ such that $g(x) \approx f^i(x)$. Then, when computing the gradient $\nabla_x f(x)$ via backpropagation, we compute a standard forward pass through $f(\cdot)$ (including the forward pass through $f^i(\cdot)$), but replace $f^i(\cdot)$ by $g(\cdot)$ on the backward pass.

The approximated gradient can then be plugged into a standard gradient-based attack. For defenses that apply some type of non-differentiable "de-noising" layer $f^1(x)$ to the input, it is often effective to approximate $f^1(x)$ with the identity function $g(x) = x$.

**EOT.**   Expectation over Transformation [AEIK18] is a standard technique for computing gradients of models with randomized components. The attack was originally proposed for situations where a randomized transformation is applied to an input $x$ before being fed into a classifier. The idea is more generally applicable for obtaining gradients of the expectation of any randomized function.

Given a randomized classifier $f_r(x)$ (where $r$ denotes the classifier's internal randomness), we can compute

$$\nabla_x \mathbb{E}_r[f_r(x)] = \mathbb{E}_r[\nabla_x f_r(x)] \approx \frac{1}{n} \sum_{i=1}^n \nabla_x f_{r_i}(x) \ ,$$

where the $r_i$ are independent draws of the function's randomness. As with BPDA, the approximated gradients can then be plugged into any standard attack.

# B   k-Winners Take All

This paper [XZZ20] proposes an activation function that is intentionally designed to mask backpropagating gradients in order to defend against gradient-based attacks.

## B.1  How The Defense Works

This defense replaces the standard ReLU activation function in a deep neural network with a discontinuous k-Winners-Take-All ($k$-WTA) function,

$$\phi_k(\mathbf{y})_j = \begin{cases} y_j, & y_j \in \{k \text{ largest elements of } \mathbf{y}\}, \\ 0, & \text{else.} \end{cases}$$

The activation function is applied to the output of a whole layer. The parameter $k$ is typically not the same for all layers. Instead, for each layer $k$ is obtained by multiplying the dimensionality $N$ of the layer's output with a constant sparsity factor $\gamma \in [0, 1]$.

The result of this particular choice of activation function is that even small changes to the input drastically change the activation patterns in the network and thus lead to large jumps in the predictions. This behavior is highlighted in the manuscript [XZZ20, Figure 5]. This chaotic partitioning of the activation patterns destroys all useful gradient information and thus prevents gradient-based attacks from finding minimal adversarial examples.

**Why we selected this defense.** There are many defenses that unintentionally make gradient descent hard. This paper takes that to the extreme and designs the defense with a "$C^0$ discontinuous function that purposely invalidates the neural network model's gradient" [XZZ20]. Because prior work has argued that this form of gradient masking is a fundamental flaw [ACW18], the existence of such a defense would go against common wisdom.

## B.2  Initial Hypotheses and Experiments

The implicit assumption of the paper is that gradient-based attacks are *strictly* stronger than black-box attacks such as score-based or decision-based attacks [CZS$^+$17, BRB18]. There are many classes of functions for which this is not true. For example, consider any undefended network and add a simple non-differentiable activation function to the final logits that quantizes them extremely finely. Then, naïve gradient-based attacks would fail because all gradients would be zero or undefined. However, estimating the gradient from finite differences would still work.[7]

In order to demonstrate that the defense is effective, the paper tests against transfer-based attacks. While it argues that "generating adversarial examples from an independently trained copy of the target network" is "the strongest black-box attack", in practice transfer attacks are often weak. The success of transfer attacks strongly depends on how close the substitute model is to the target model. Training with the same architecture is insufficient; networks trained from different initializations can be far apart in their internal representations. The perturbation size necessary for transfer attacks to succeed is thus often significantly larger than for white-box attacks [PMG16, TKP$^+$18]. One setting where transfer-based attacks can be strong is if the model $f$ can be transformed into a nearly identical model $f'$ for which it is easier to generate adversarial examples (e.g., the BPDA attack of [ACW18] can be seen as an instance of this approach). In contrast, even the most restricted class of direct attacks, decision-based attacks (which only use the final label decision of the model [BRB18]), can perform as well as gradient-based attacks if the attack is run until convergence.

While the omission of score- and decision-based attacks is the biggest concern, a few other issues make it difficult to assess the true robustness of this defense from the initial evaluation. First, the lack of a distortion-accuracy plot makes it difficult to spot gradient-masking issues [CAP$^+$19]. Second, some attacks (C&W, DeepFool) are actually optimizing the $\ell_2$ metric and not the $\ell_\infty$ metric under which the accuracy is measured, meaning that the values for those attacks should be much higher than for proper $\ell_\infty$ attacks (but they are often close, within a few percentage points). Finally, the attacks in the paper are only applied with a small number of steps (20-40), which might have prevented attacks from converging, and without any repetitions (which can often reduce accuracy by 10% or more).

For our experiments, we evaluate the authors' released defense on CIFAR-10 with $\ell_\infty$ bounded attacks of distortion $\epsilon = 8/255$. We evaluate $k$-WTA at sparsity ratio $\gamma = 0.1$ as the paper reports the

Figure 1: Loss landscape of the k-Winners Take All defense. The plot shows the difference between the logit of the ground-truth class and the next highest logit along a random perturbation direction $x + \epsilon \operatorname{sign}(\delta)$. Note the x-axis scale: the largest perturbation changes the input by 0.1%.

strongest results with this setting. We evaluate both a vanilla and an adversarially trained ResNet-18 for which the paper claims $13.1\%$ and $50.0\%$ accuracy (respectively). All our experiments rely on implementations in Foolbox [RBB17].

We begin our evaluation by running a standard PGD attack [MMS$^+$17], with 400 steps instead of 40 as performed in the paper's original evaluation. We find that over the progression of the attack, the cross-entropy loss exhibits high fluctuations. In fact, adversarial examples were seemingly found by chance and the attack did not reliably decrease the average loss. This suggests that the extreme jagged loss surface of the $k$-WTA-model could even *decrease* the accuracy compared to a baseline model: because of the high fluctuations in the logits, it is possible that even in correct and high-confidence regions of the original model, the fluctuations in the logits could introduce adversarial examples.

We then looked at the loss function close to clean samples. As is visible in Fig. 1, even very small perturbations can flip the loss to a new state. The loss stays roughly constant until it flips again if the perturbation is further increased. This explains why standard gradient-based attacks fail: the gradients can only highlight the direction of descent on each plateau but cannot take into account the substantial non-linear activation shifts at the boundary of the plateaus. In other words, the gradients are only valid in an extremely small and confined region but are not representative of the descent directions over larger regions. Appendix F contains another defense that fails for a similar reason.

We therefore hypothesize that a strong black-box attack that directly queries the target model and estimates the gradients from finite differences over a larger space would be an effective attack on this defense.

### B.3 Final Robustness Evaluation

Our final attack strategy was to average out the randomness by estimating the average local gradient $\delta$ at point $x$ with $M = 20000$ random perturbations drawn from a standard normal distribution with standard deviation $\epsilon = 8/255$, i.e.,

$$g(x) = \frac{2}{\sigma M} \sum_{j=0}^{M/2} \left[ \nabla_x L_{CE}(f(x + \delta), y) + \nabla_x L_{CE}(f(x - \delta), y) \right]$$

where $\delta \sim \mathcal{N}(\mu = 0, \sigma^2)$ and $L_{CE}$ is the cross-entropy loss. We run this attack for 100 steps and successfully generated adversarial examples for both the vanilla as well as the adversarially trained ResNet-18 that reach 0% and 0.16% accuracy respectively.

Observe that this confirms our concerns: not only is k-WTA not an effective defense, it makes adversarial training *worse*. This is not the first time that this has happened: [ACW18] previously observed the same effect of a defense *reducing* the efficacy of adversarial training.

**Lessons Learned.**

1. Score-based or decision-based attacks can work as well or even better than gradient-based attacks, in particular when the gradients are (intentionally) masked.
2. Transfer-based attacks, while cheap to implement, require a highly similar source model. The best transfer attacks typically succeed less than half of the time, and thus transfer attacks typically cannot disprove robustness claims less than roughly $50\%$.

# C The Odds are Odd

This paper [RKH19] proposes a statistical test for detecting adversarial examples, based on the distribution of a classifier's logit values (i.e., the per-class scores in the last layer of the classifier, before the softmax computation).

## C.1 How The Defense Works

The main assumption that motivates the paper's defense is that adversarial examples are less robust to noise than benign examples. More specifically, for a given input $x$, the defense compares the "clean" logit vector $z(x)$ to a "noisy" logit vector $z(x + \delta)$, where $\delta \sim \mathcal{N}$ is sampled at test-time from some fixed distribution $\mathcal{N}$ (e.g., a multinomial Gaussian). The assumption is that for clean examples, $z(x) \approx z(x + \delta)$, while for adversarial examples the two logit vectors will differ significantly.

In more detail, the defense looks at variations in the difference between the logit of the predicted class $y$ and the logits of other classes $i \neq y$. Let

$$\Delta_{y,i}(x) = z(x)_i - z(x)_y \ ,$$

where $y$ is the predicted class, $i \neq y$ is some other class, and $z(x)_j$ is the logit for class $j$. Note that $\Delta_{y,i}(x)$ is always negative. For each class, the defense checks whether these logit differences are robust to noise by computing[8]

$$\bar{\Delta}_{y,i}(x) = \mathbb{E}_{\delta \sim \mathcal{N}}\left[\Delta_{y,i}(x + \delta) - \Delta_{y,i}(x)\right] \ .$$

In practice, the expectation is approximated by sampling $k$ independent noise values $\delta \sim \mathcal{N}$. An input is rejected as adversarial if for any class $i$ other than the predicted class $y = f(x)$, we have $\bar{\Delta}_{y,i}(x) > \tau_{y,i}$, where the $\tau_{y,i}$ are thresholds set to achieve some fixed false-positive rate.

**Why we selected this defense.** The approach taken by this defense is very simple, but relies on a common misconception that the peculiarities of adversarial examples created by standard attacks can be used to detect *all* adversarial examples. This defense is very similar in spirit to the one analyzed in Appendix N.[9]

## C.2 Initial Hypotheses and Experiments

The paper's evaluation shows that existing attacks such as PGD [MMS+17] or C&W [CW17b] indeed produce adversarial examples with large values of $\bar{\Delta}_{y,i}(x)$. Due to the stochastic nature of the defense, the defense is further evaluated on an attack that combines PGD with "Expectation Over Transformation" (EOT) [AEIK18] to produce adversarial examples robust to noise from $\mathcal{N}$.[10] While more effective, this attack's success rate remains low.

The paper's evaluation shows no clear "red-flags", in part because very few results beyond the absolute detection numbers are reported (which perhaps could be considered a red-flag: it is helpful for defenses to give additional statistics if only to diagnose potential errors).

To better understand why existing attacks fail, and what a successful attack would have to achieve, we take a closer look at the detection statistics computed by the defense:

$$\begin{aligned}
\bar{\Delta}_{y,i}(x) &= \mathbb{E}_{\delta \sim \mathcal{N}}\left[\Delta_{y,i}(x + \delta) - \Delta_{y,i}(x)\right] \\
&= \mathbb{E}_{\delta \sim \mathcal{N}}\left[(z(x + \delta)_i - z(x + \delta)_y) - (z(x)_i - z(x)_y)\right] \\
&= \underbrace{\mathbb{E}_{\delta \sim \mathcal{N}}\left[(z(x + \delta)_i - z(x + \delta)_y)\right]}_{s_1} + \underbrace{(z(x)_y - z(x)_i)}_{s_2} \ .
\end{aligned}$$

We focus on the scores $s_1$ and $s_2$ independently. Observe that $s_1 + s_2$ must be small to evade the defense.

The term $s_1$ is positive and large if the model often switches its prediction to some other class when the input is perturbed with noise. The term $s_2$ is positive and large if the model is overly confident in its original prediction of class $y$, compared to other classes $i \neq y$.

From here, we notice an issue with the type of attacks evaluated in the paper. PGD creates examples with abnormally large values of $s_2$ (compared to benign examples) as this is precisely the objective that the attack maximizes. In contrast, attacks such as C&W that find the smallest adversarial perturbation will find examples that lie very close to the decision boundary. For these, $s_2$ will be small, but $s_1$ will be large as examples close to the boundary are typically not robust to noise. Finally, combining PGD with EOT will reduce the term $s_1$, but will not prevent the attack from finding examples with abnormally large values of $s_2$.

Thus, a successful attack will be one that finds adversarial examples for which: (1) the model's robustness to noise is no smaller than for an average benign example; and (2) the model's confidence is no larger than for an average benign example. Interestingly, for a logistic regression model these two properties of resistance to noise and model confidence are directly related: the model's confidence depends on an input's distance to the decision boundary, which in turn defines the input's robustness to (Gaussian) noise.

## C.3  Final Robustness Evaluation

We perform our evaluation on CIFAR-10, using the code released by the authors. We consider $\ell_\infty$ attacks bounded by $\epsilon = 8/255$, for which the paper claims a $71.4\%$ detection rate with a standard wide ResNet architecture. We consider two default settings proposed by the authors, one where the noise $\delta$ is sampled from a single multinomial Gaussian, and one where the terms $\bar{\Delta}_{y,i}(x)$ are computed for 10 different sources of noise (Gaussian and uniform distributions with different variances). In the latter case, inference is extremely expensive: classifying a single example $x$ requires computing $z(x + \delta)$ for thousands of noise values $\delta$.

As a first attack strategy, we create adversarial examples for which the term $s_2$ is the same order of magnitude as for benign samples. To do this we generate an adversarial example that targets the logits of an arbitrary benign example from another class. This is a "feature-level" attack [SCFF16], where we aim to match the model's internal representation of another class, rather than just changing the predicted output. For each class $t$, we pick a benign example $x_t$ of that class and compute the logits $z(x_t)$. Then, for an example $x$ of class $y$, we pick a target $t \neq y$ and use PGD to create adversarial examples $x'$ that minimize the objective:

$$||z(x') - z(x_t)||_2^2 . \tag{1}$$

This attack is surprisingly effective. The attack always fools the classifier, and only a small number of adversarial examples are detected due to abnormally large values $\bar{\Delta}_{y,i}(x)$. For the defense that uses a single noise source, $17\%$ of attacks are detected. For the defense that combines 10 noise sources (and therefore needs to lower the thresholds $\tau_{y,i}$ to achieve a fixed false-positive rate), only $10\%$ of attacks are detected.

To increase the attack strength, we further add the $s_1$ term to our objective. To minimize $s_1$, we apply standard EOT. Crucially, we apply EOT to the logit-matching objective in equation (1), rather than to the standard PGD objective. The resulting attack always fools the classifier maintaining a $0\%$ detection rate; lower than the false positive rate.

A very similar attack on this defense was independently presented by [HKP19].

**Lessons Learned.**

1. Most existing attacks produce adversarial examples that are either abnormally close to the decision boundary (e.g., C&W), or have abnormally high confidence (e.g., PGD). While this makes these attacks easy to detect, an adaptive attack can create examples that are similarly close to boundaries and have similar confidence as benign examples.
2. This defense serves as a canonical example of the power of a very simple "logit-matching" attack that is effective against multiple defenses.
3. When attacking a randomized defense, expectation-over-transformation (EOT) is an insufficient strategy if the attack *objective* is not also appropriately adapted.

---

**Algorithm 1** Generative Classifier [LBS19].

---

**Input**: Data point $x$
**Output**: Logits $z(x)$

---

$\textbf{for } k \in [1, K] \textbf{ do}$           `// iterate over class labels`
    $\vec{\mu}, \vec{\sigma} = \texttt{enc}(x, k)$        `// class-conditional encoder`
    $\textbf{for } i \in [1, N] \textbf{ do}$
        $\eta \leftarrow \mathcal{N}(\vec{\mu}, \vec{\sigma} \cdot I)$        `// sample latent vector`
        $x^* = \texttt{dec}_1(\eta)$        `// decode to input`
        $y^* = \texttt{dec}_2(\eta)$        `// decode to logits`
        $L_{\text{recons}} = ||x - x^*||_2^2$        `// input loss`
        $L_{\text{CE}} = \text{cross-entropy}(y^*, k)$        `// label loss`
        $\text{p}_{\text{prior}} = \log(\mathcal{N}(\eta; 0, I))$        `// log prior`
        $\text{p}_{\text{posterior}} = \log(\mathcal{N}(\eta; \vec{\mu}, \vec{\sigma} \cdot I))$        `// log posterior`
        $\texttt{score}_i = -L_{\text{recons}} - L_{\text{CE}} + \text{p}_{\text{prior}} - \text{p}_{\text{posterior}}$
    $\textbf{end}$
    $l_k = \log\left(\frac{1}{N} \sum_{i=1}^{N} \exp(\texttt{score}_i)\right)$        `// logit`
$\textbf{end}$
$\textbf{return } z(x) = [l_1, \ldots, l_K]$

---

# D  Are Generative Classifiers More Robust?

This defense [LBS19] is fairly complex and displays many characteristics of what makes an evaluation challenging, such as the use of multiple models, aggregation of multiple losses, stochasticity, and an extra detection step.

## D.1  How The Defense Works

The defense is based on the Variational Auto-Encoder framework [KW14]. Specifically, the authors assume that data points $(x, y)$ are generated by some unknown process involving an unobserved latent variable $\eta$.[11] The paper explores different ways of factoring the joint distribution $p(x, y, \eta)$. We focus on the "GBZ" model that the authors found to be most robust (see [LBS19] for a graphical illustration):

$$p(x, y, \eta) = p(\eta)p(y|\eta)p(x|\eta) .$$

Rather than delving into the theory of Variational Auto-Encoders, we present here the high-level classification algorithm implemented in the source code released by the authors (see Algorithm 1). This mirrors our own mental model when assessing the robustness of this defense: understanding and evaluating what the implementation does is often more useful than understanding the theoretical or intuitive explanations made in the paper.

The classifier combines three models, a per-class encoder `enc` that produces parameters for sampling random latent vectors $\eta$, and two decoders `dec`$_1$, `dec`$_2$ that reconstruct the input and label from the latent vectors. All these models use standard CNN architectures. For each class, the defense runs these three models for $N$ random latent vectors ($N = 10$ by default), and computes four scores that combine to form the logit for that class.

The paper proposes two extensions. First, to scale to CIFAR-10, the input $x$ is fed into a feature-extractor $\phi = \Phi(x)$, where $\Phi$ is an intermediate layer of a pre-trained VGG classifier. Second, the paper adds a detection step, that rejects inputs with "unusual" class probabilities $f(x)$ (obtained by applying the softmax function to the logits $z(x)$). We focus on the best-performing approach, where the detector rejects inputs with high KL-divergence between $f(x)$ and the mean probability vectors from training inputs of the same class.

**Why we selected this defense.**  The defense relies on a generative approach which has often been postulated as being more robust than purely discriminative models. This defense's complexity makes

it particularly hard to evaluate and illustrates the advantage of focusing on simple attacks that precisely target the defense's most important pieces.

## D.2   Initial Hypotheses and Experiments

Our initial observation is simply that this defense is *complex*[12] (see Appendix G for another complex defense). It involves three networks, four scoring terms, many mixes of exponentials and logarithms, heavy randomization, and a detection step. It thus comes at little surprise that existing attacks (e.g., PGD [MMS$^+$17] or C&W [CW17b]) are shown to be ineffective in the paper: combining multiple loss terms effectively is never easy. The evaluation explicitly demonstrates the insufficiency of these attacks: attacks with unreasonably large $\ell_\infty$-bounds ($\epsilon = 0.5$ on MNIST, and $\epsilon = 25/255$ on CIFAR-10) fail to bring the classifier's accuracy to $0\%$.

We set out to test the following initial hypotheses, informed by our reading of the paper and of the released code:

1. It is unlikely that the four terms that make up $\text{score}_i$ are all of similar importance. Targeting a single one might be sufficient and easier to optimize.

2. Given a successful attack on the classifier, it is plausible that only slight tweaks to the distribution of the logits $z(x)$ will be needed to also fool the detector.

We evaluate the paper's MNIST and CIFAR-10 models, using code and models provided by the authors. For simplicity, we exclusively consider a standard $\ell_\infty$ threat-model with $\epsilon = 0.3$ on MNIST and $\epsilon = 8/255$ on CIFAR-10 (the paper also gives results for other bounds and for $\ell_2$ attacks).

After a few standard yet ineffective tests (increasing PGD iterations and the number of random samples $N$), we tried a simple attack using random perturbations and many restarts. This attack was surprisingly effective: after a few thousand trials the MNIST model's accuracy dropped below $50\%$. While trying to build a stronger attack (see details below), we noticed a curious fact about this defense: the success of our random attack is not due to the model's lack of noise-robustness, but to the high variance in the model's estimate of the $\text{score}_i$ values. Indeed, even when classifying an *unperturbed* input drawn from the test set multiple times in a row, the model misclassifies it with some small ($5\%$-$10\%$) probability. The defense is thus vulnerable to a "trivial" attack that continuously submits the same input until the model misclassifies it by chance. This can be fixed by increasing the number $N$ of sampled latent vectors, at a cost of more expensive inference.

We thus searched for an attack that could produce more "robust" adversarial examples. To test our first hypothesis, we compared the different terms in the scores $\text{score}_y$ and $\text{score}_i$, where $y$ is the true class and $i \neq y$ some other class. We found that $L_{\text{CE}}$ is the only important term. The other terms vary little across class scores. To generate more robust attack samples, we also increase the defense's parameter $K$, the number of randomly sampled latent vectors from 10 to 100 (this is essentially the EOT technique [AEIK18, ACW18]). The created samples are tested on a defense with the original parameter setting of $K = 10$. With a standard PGD attack to maximize the $L_{\text{CE}}$ term for the true class, we reduce the classifier's accuracy below $5\%$ on both MNIST and CIFAR-10. These adversarial examples are also robust to the model's high variance.

We found that despite only targeting the classifier, this attack is moderately successful against the KL-divergence detector: only $20\%$ of the MNIST attacks and $10\%$ of the CIFAR-10 attacks are detected. This gives credence to our second hypothesis. Finally, we proceed to find adaptive strategies that fool both the classifier and detector.

## D.3   Final Robustness Evaluation

We first directly minimized the KL-divergence, which proved computationally expensive (as it requires optimizing over all class scores) and unsuccessful. We further tried to align the model's decoded logits $y^* = \text{dec}_2(\eta)$ to those produced on a clean input from a different class, using the "feature-adversary" [SCFF16]. The aim is to match the model's internal representations to those

obtained with another input, so that downstream statistics (e.g., the softmax output) match as well. On CIFAR-10, this attack achieves 99% success against the combined classifier and detector.

The insight of feature adversaries suggests a much simpler attack we had overlooked. As the CIFAR-10 classifier is of the form $f(\Phi(x))$ where $\Phi(x)$ are features extracted from a VGG model, an attack that transforms these features to those of a different class will be successful. Concretely, we find an adversarial example $x'$ that minimizes $L(x') = ||\Phi(x') - \Phi(x_t)||_2^2$, where $\Phi(x_t)$ are the VGG features for a clean example $x_t$ of a different class. This attack succeeds with 100% probability.

For the MNIST classifier, the same attacks as above produce adversarial examples that each succeed with 90% probability (over the model's internal randomness). This limitation may be inherent as the model also has a 10% failure rate when classifying a clean input. To verify that our attack's "failure" is indeed due to the classifier's excessive variance, we made the sampling of latent vectors $\eta$ deterministic (i.e., given parameters $\vec{\mu}, \vec{\sigma}$, the samples $\eta_1, \ldots, \eta_N$ are produced using a fixed seed). In this simplified setting (which has no effect on the model's accuracy), we reach 100% success rate with the following PGD attack: we maximize $L_{\text{CE}}$ for the true class, and stop as soon as we find that the perturbation computed by the current PGD iteration also fools the detector.

**Lessons Learned.**

1. For complex defenses, it is useful to decompose the contributions of individual classification steps, in order to determine which ones should be focused on.

2. Evaluating a defense on random noise can reveal surprising failure cases.

3. Feature adversaries are useful for jointly evading a classifier and detector, as they ensure that any statistics computed on top of adversarial features match those from clean examples.

# E    Robust Sparse Fourier Transform

This paper [BMV18] introduces a defense to $\ell_0$ adversarial examples through what is called a "robust sparse Fourier transform".

## E.1    How The Defense Works

For a given input, this paper proposes to defend against $\ell_0$ adversarial examples by taking each image, "compressing" it by projecting to the top-$k$ coefficients of the discrete cosine transform, inverting that to recover an approximate image, and then classifying the recovered image. The classifier is trained on images compressed in this way so that it remains accurate.

In more detail, this paper uses the Iterative Hard Thresholding (IHT) method which consists of $T$ iterations, each of which performs a pass at compressing the input $x$ with a Fourier transform and then recovering the inverted input. This defense's idea is conceptually similar to that of the ME-Net defense studied in Section L, although ME-Net has the additional challenge of being randomized.

**Why we selected this defense.**    There are not many defenses to $\ell_0$ adversarial examples, and applying a Fourier transform intuitively makes sense as a possible defense.

## E.2    Initial Hypotheses and Experiments

This paper does not contain an analysis of its robustness to adaptive attacks. Instead, it demonstrates that adversarial examples on a base classifier, when pre-processed by the defense method, are no longer classified as adversarial by the base classifier afterwards. This is insufficient to demonstrate robustness as has been argued extensively [CAP+19].

In particular, the paper claims over 75% accuracy at an $\ell_0$ robustness of 55 on MNIST; however, it has been shown that an $\ell_0$ distortion of 25 is sufficient to actually cause humans to change their classification of MNIST images [JBC+19]: it is therefore improbable that any defense could achieve a stronger result.

We suspected that an adaptive attack would break this defense without any other difficulty. The paper argues robustness on multiple $\ell_0$ attacks, including the one presented by [CW17b] which often has lower distortion than the others considered. We therefore adapt this attack to the defense.

### E.3 Final Robustness Evaluation

To begin we implement this defense as a differentiable pre-processor in front of the neural network classifier (using code provided by the authors). Implementing the procedure as a differentiable function requires a small amount of work as there are no non-differentiable components: everything is already fully differentiable. We then apply the $\ell_0$ attack from [CW17b] on this combined function without any further modification. We run the attack with a maximum of $500$ iterations of gradient descent before deciding which pixels to freeze at their original value. Instead of running a multi-targeted attack we run the attack in an untargeted manner, resulting in a larger distortion than we could otherwise achieve but $10\times$ faster and mirroring the original evaluation. We find the resulting attack is successful at generating adversarial examples with a median distortion of $14.8$ pixels, within the margin of error of the $15$ pixels reported by [CW17b] for an undefended model.

**Lessons Learned.**

1. It is important to optimize the combined loss function on $f(p(x))$ whenever defenses propose a pre-processing function $p$.
2. Not all pre-processing functions are hard to differentiate. Some just require to be implemented in a library that supports auto-differentiation.

## F Rethinking Softmax Cross Entropy

This paper [PXD+20] proposes a new loss function to use during training in order to increase adversarial robustness.

### F.1 How The Defense Works

The paper trains a feature-extractor $g$. Note that for this classifier, the output features $g(x) \in \mathbb{R}^N$ do not correspond to class-scores. Instead of training models with standard softmax cross entropy, this paper proposes the *Max-Mahalanobis center (MMC) loss*, defined as

$$L_{\text{MMC}}(g(x), y) = \frac{1}{2}\|g(x) - \mu_y\|_2 .$$

Here, the $\mu_y \in \mathbb{R}^N$ are the centroids of the Max-Mahalanobis distribution; the exact choice of $\mu$ is not important for understanding the defense. Given the trained feature-extraction network function $g$, the network classifies an input $x$ into one of $K$ classes as

$$f(x) = \arg\min_{1 \leq i \leq K}\|g(x) - \mu_i\|_2.$$

We comment on one further observation we make of this paper. While not a failure mode, this paper contains a large amount of theory and proofs of various facts that are mostly unrelated to the actual defense proposal itself. While this theory may be interesting in its own right, it is not always necessary to follow the theory for *why* the defense was proposed in order to still evaluate the proposed defense.

**Why we selected this defense.** There are a number of defenses that aim to replace the standard softmax cross entropy loss with alternate functions. We select this defense as as a representative example of this style of defense because it is a strengthening of a prior defense to adversarial examples [PDZ18], and as such we expected it would be a strong candidate.

### F.2 Initial Hypotheses and Experiments

This defense does not perform an analysis of robustness to adaptive attacks; as such, we expect that carefully constructing a loss function that is well suited to the defense will allow us to reduce the accuracy significantly.

In general, whenever a defense changes the loss function used to train a model, our first hypothesis is always that the new model has a loss surface that is not well suited to attacks with the standard softmax cross entropy loss. Towards that end, we perform the analysis that we would have liked to see in the original paper.

The original evaluation uses standard attacks such as PGD that maximize the cross-entropy loss over the model's predictions. The paper defines the defense's logit vector $z(x)$ as $z(x)_i = -\|g(x) - \mu_i\|_2$ for $1 \le i \le K$, and then applies the softmax function to $z(x)$ to obtain class probabilities. Initially we examined the value of the logits of the classifier and found that they were extremely abnormal compared to a typical classifier. The largest logit tended to be near $0$ and the remaining logits near $-200$. This reminded us of distillation as a defense [PMW$^+$16], which has a similar effect on the logits. A simple method for breaking distillation is to simply divide the logits by a large constant [CW16]; however when we tried this it was not sufficient to break the defense completely (although it did help some and reduced the classifier accuracy on CIFAR-10 from $24\%$ to $18\%$. That this alone allowed the attacks to succeed more often was worrying: shifting the logits by a constant should not improve the efficacy of attacks.

As such, we hypothesized that an attack would need to design a loss function that better captured what the defense was doing in order to succeed more often.

### F.3 Final Robustness Evaluation

The justification for what the defense does, as well as its training procedure, are complicated. Yet, inference is simple. Given the trained feature-extraction network function $g$, the network assigns the label

$$\arg \min_i \|g(x) - \mu_i\|_2.$$

That is, there are $K$ (for a $K$-class classification problem) different "target" vectors $\{\mu_i\}_{i=1}^K$ and the classifier returns whichever target is closest given the feature vector $g(x)$. In this sense, this defense is similar to the error correcting defense from Section G.

This gives us our candidate attack procedure. Instead of indirectly performing softmax cross entropy loss on the distances to $\mu$, we directly target each of the target vectors $\mu$. That is, we directly optimize the loss function

$$L(x) = \|g(x) - \mu_i\|_2^2$$

for each target class $i$, again performing a multi-targeted attack. Directly optimizing the loss function in this way reduces the classifier accuracy to under $0.5\%$ at distortion $\epsilon = 0.031$ on CIFAR-10.

**Lessons Learned.**

1. Any time a defense changes the loss function used during training, the loss function used to attack should always be studied carefully.

2. The softmax cross-entropy loss is hard to optimize when there are large differences in the model's logit values.

## G Error Correcting Codes

This paper [VS19] proposes a method for training an ensemble of models with sufficient diversity that the redundancy can act as error correcting codes to allow for robust classification.

### G.1 How The Defense Works

Given an $K$-class classification problem, this paper proposes to generate a matrix $A \in \{0, 1\}^{K \times M}$ of binary codewords.[13] The defense trains $M$ different classifiers so that classifier $f_i$ outputs a value between $0$ and $1$ with the objective that $f_i(x) = A_{j,i}$ when the label for $x$ is $j$. This way, each classifier is performing a binary task prediction where each class is randomly assigned to either the $0$ or the $1$ class. Clearly if there are not enough classifiers, we can not uniquely recover the result

given the output of the models (in particular, we must at least have that $M \geq \log(K)$). Thus, each classifier $f_i$ is trained on a slightly different task, aiming to increase diversity of the classifiers so that adversarial examples will not transfer between the classifiers.

Specifically, each classifier is trained as a function $z_i \colon \mathcal{X} \to \mathbb{R}$ and then $f_i(x) = \text{sigmoid}(z_i(x))$. To generate the final predictions, define the vector $Z(x) = [z_1(x) \quad z_1(x) \quad \ldots \quad z_M(x)]$, and then return $f(x) = \arg\max_{1 \leq j \leq K} \|\text{sigmoid}(Z(x)) - A_j\|_2$.

**Why we selected this defense.** We study this defense, along with the the Ensemble Diversity [PXD$^+$19] defense in the prior section, as representative examples of defenses that ensemble together multiple models. Because the prior defense was not robust, we selected another ensembling defense to test if it would be any more robust to adversarial examples.

## G.2 Initial Hypotheses and Experiments

This is another defense that we would classify as *complex*, even though it may not look that way initially. It is not *difficult*—it is an easy defense to understand, conceptually—but there are several moving pieces where multiple classifiers are trained independently and each classifier is trained on a different task. As we saw previously with the generative model defense [LBS19] in Appendix D, attacks on complex defenses do not have to be complex once the weak links are identified.

Our main concern with this paper was that it reports nearly $40\%$ accuracy at an $\ell_\infty$ distortion of $\varepsilon = 0.5$ which is a tell-tale sign of gradient masking [ACW18]: any defense at this distortion *must* be breaking the gradient descent process somehow. This paper draws exactly the opposite conclusion from this figure, stating that "Because model accuracy rapidly drops to near $0$ as $\varepsilon$ grows, the [accuracy versus distortion] figure provides crucial evidence that our approach has genuine robustness to adversarial attack and is not relying on "gradient-masking"." While it is true that the accuracy does go down, it does not reach $0\%$ accuracy at $\varepsilon = 0.5$ and therefore the defense **is** causing gradient masking.

When we study the code we find the reason for this. Each of the classifiers $f_i$ are followed by a sigmoid activation function in order to map to the correct output space for the 0-1 codes. However, after this, to compute the unified predictions, the implementation takes the $\log$ of the result and then feeds this into the softmax function. Finally, to generate adversarial examples with the cross-entropy loss, another $\log$ is required. This is exceptionally numerically unstable and we suspected it was the cause of the gradient problems.

## G.3 Final Robustness Evaluation

Given the $M$ neural networks $\{f_i\}_{i=1}^M$, we remove the final sigmoid layer after each model, and remove the $\log$ operation in merging together the predictions. In order to be more robust to further numerical instabilities, we replace the standard softmax cross entropy with the hinge loss proposed by [CW17b]. This attack is able to substantially degrade the model accuracy to lower than $20\%$ on CIFAR-10 at $\varepsilon = 0.031$, however it does not completely reduce it to zero.

We then apply a few tricks that are typically able to reduce the accuracy by a few percentage points more. First, we run a multi-targeted attack and generate untargeted adversarial examples by targeting each of the other 9 target labels. Second, we run each of these multi-targeted attacks five times, with different initial random steps for the PGD attack. Finally, if *any* inner iteration of PGD succeeds at generating an adversarial example we take this; we don't require that the *final* output of PGD after exactly 100 steps is an adversarial example. This finally brings the accuracy of the classifier to under $5\%$ at $\varepsilon = 0.031$, compared to the reported $57\%$ accuracy at this distortion.

**Lessons Learned.**

1. Combining together multiple neural networks into a single combined defended network is extremely error-prone; however, once a simple method is identified, it often is effective.

2. There are number of tricks that can increase attack success rate by a few percentage points each; when model accuracy is already low, these tricks can be sufficient to reduce accuracy to near-zero.

# H  Ensemble Diversity

This paper [PXD⁺19] proposes to train an ensemble of models with an additional regularization term that encourages diversity. Compared to single models or less diverse ensembles, this additional diversity is supposed to make it more difficult for attacks to find minimal adversarial examples.

## H.1  How The Defense Works

Let $f_m(x)$ be the probability vector of the $m$-th model in the ensemble and $f(x) = \sum_m f_m(x)$ be the probability vector output by the ensemble. The ensemble of models is trained on the following objective,

$$L(x, y) = -\alpha \mathcal{H}(f(x)) - \beta \operatorname{Vol}^2 \left( \left\{ f_m^{\backslash y}(x) \right\} \right) + \sum_m^M L_{CE}(f_m(x), y)$$

where $\mathcal{H}(\cdot)$ is the Shannon entropy, $L_{CE}$ is the standard cross-entropy loss applied to the prediction of each model and the middle term is the volume spanned by the probability vectors of the individual models of the ensemble (each normalized to unit length). Note that the volume is only computed over the non-maximal predictions, i.e. we remove the leading class from each probability vector $f_m(x)$. The rational for doing this is that this will balance accuracy and diversity (the volume can only increase if the predictions in the non-maximal classes are diverse). The weighting coefficients $\alpha, \beta$ are hyper-parameters.

**Why we selected this defense.**  There is a long line of work studying the robustness of neural network ensembles. Most other work in this direction has not succeeded [HWC⁺17]. We study this defense, along with that of [VS19] (Appendix G) and [PXD⁺19] (Appendix I), to evaluate what appears to be the strongest defenses in this class.

## H.2  Initial Hypotheses and Experiments

None of the terms in the loss function seem to encourage any kind of gradient masking. There is no other mechanism other than the change of the training objective, thus suggesting that standard off-the-shelve gradient-based attacks should be successful.

The paper's results section might impress with the large number of attacks employed against the ensemble model. However, BIM [KGB16], PGD [MMS⁺17] and MIM [DLP⁺18] are extremely similar and if one of these attacks fails the others are likely to fail as well. Instead, it is better to employ a more diverse set of attacks, e.g. by including the score-based pixel-wise attack [SRBB19] or the Brendel & Bethge (B&B) attack [BRM⁺19] (the latter of which was not available at the time of initial publication). Similarly, the results for FGSM are only useful in that they might signal gradient masking (in particular if FGSM finds adversarial perturbations where multi-step methods do not) but in general they should always perform worse then more powerful attacks (which they do here).

Nonetheless, the paper's results contains a few odd values. For one, the diverse ensemble reaches higher robustness on CIFAR-10 than a state-of-the-art adversarially trained model (30.4% compared to 27.8% for $\epsilon = 0.02$). That is fairly unlikely and not mentioned anywhere in the main text. Second, it is odd that the BIM attack tends to be more effective than PGD given that the two only differ in their starting points (BIM starts from the original sample while PGD starts from a random point within the allowed $\ell_\infty$ norm ball). For adversarial training, choosing a random starting point can mitigate some gradient masking issues. In general, choosing a random starting point makes the attack converge to different perturbations which can increase success rate by 10% or more if each sample is attacked multiple times.

These oddities suggest that the attacks might not have converged. And indeed, the paper reports to have used only 10 iterations for BIM, PGD and MIM (with a step size of $\epsilon/10$). Such a small number of iterations is unlikely to allow the attacks to converge and to find the strongest adversarial perturbation, suggesting that the attack success can be substantially increased simply by increasing the number of iterations. The same observation holds for the C&W [CW17b] and EAD [CSZ⁺18] attacks for which the paper reports 1,000 iterations. Due to the inner hyper-parameter optimization that both attacks employ, this number is still on the low side.

Taken together, our hypothesis was that simply increasing the number of iterations and/or combining PGD with a substantially different attack such as B&B can substantially increase attack success.

### H.3    Final Robustness Evaluation

For our experiments, we evaluate the authors' released defense on CIFAR-10 with $\ell_\infty$-bounded attacks of distortion $\epsilon = 0.01$. We evaluate the strongest ensemble trained with $\alpha = 2$ and $\beta = 0.5$ for which the paper reports an accuracy of 48.4%. Each model of the ensemble is based on the ResNet-20 architecture.

We started from the evaluation code as well as the pre-trained model weights provided by the authors. Simply increasing the step size by a factor of three reduces the accuracy from 48% to 26%. Increasing the number of iterations to 50 decreases the accuracy to 20% while 250 iterations reduced the accuracy to 10%. A further increase did not change the result, suggesting that 250 iterations are sufficient for convergence.

We then repeated the attack three times for each example, which decreased accuracy to 7% for 250 iterations. The remaining samples seemed to be difficult for PGD, which is why we chose the $\ell_\infty$ version of the B&B attack [BRM$^+$19]. In contrast to PGD, which starts at a point close to the clean sample and moves outwards to the worst-case adversarial within the $\epsilon$-ball, the B&B attack starts from a misclassified sample far away from the clean sample and then moves along the boundary of adversarial and non-adversarial images towards the clean sample. As initial points for the B&B attack, we chose large-perturbation adversarial examples ($\epsilon = 0.15$) generated by PGD with 20 steps. We then applied B&B with 20 steps. This was sufficient to reach 5% accuracy. Repeating the attack a few times on the remaining samples reduced the accuracy to 0%.

**Lessons Learned.**

1. It is important to make sure that attacks actually converged and that the right hyper-parameters have been chosen. Also, if an attack uses random starting points, the attack should be repeated several times on each sample.

2. Using very similar attacks such as BIM, PGD or MIM is unlikely to yield very different results. Instead, attacks that use substantially different strategies such as the B&B attacks [BRM$^+$19] should be considered.

## I    EMPIR

This paper [SRR20] is the third defense we study that develops a method to defend against adversarial examples by constructing an ensemble of models, but this time with mixed precision of weights and activations.

### I.1    How The Defense Works

This paper trains multiple models $\{f_i\}$ with different levels of precision on the weights and on the activations for each and returns the output as the majority vote of the models. Concretely, for a CIFAR-10 defense the paper uses one full-precision model, one model trained with 2-bit activations and 4-bit weights, and another model with 2-bit activations and 2-bit weights.

**Why we selected this defense.**    The prior two ensemble defenses we studied were not effective; however this defense takes a different approach by training low-precision models.

### I.2    Initial Hypotheses and Experiments

This paper does not contain an explicit analysis of its robustness to adaptive attacks, but performs an analysis of the defense to standard gradient-based attacks on the ensemble. However, given that the defense has potentially non-differentiable layers we believed that it may be possible for various forms of gradient masking to occur. Investigating the code, we found that the paper *does* include a BPDA-style estimate of the gradient of the backward pass. Thus, we expected that performing

gradient descent with respect to the entire model ensemble and carefully ensuring that gradient information was correctly propagated through the model would suffice to bypass this defense.

### I.3   Final Robustness Evaluation

We begin by constructing a loss function that we will use to generate adversarial examples. As a first attempt at an attack, we form the simplest loss function that we could imagine: take the class probability vectors of the three models $f_1(x), f_2(x), f_3(x)$ and average them component-wise so that the prediction of our agglomerated model for class $i$ is given by $\hat{f}(x)_i = \frac{1}{3}(f_1(x)_i + f_2(x)_i + f_3(x)_i)$. Then we perform PGD on the cross-entropy loss for this model $\hat{f}$. Notice that this loss function *is not consistent*, violating one of our lessons: because the classifier takes the majority vote in order to decide the final prediction, only two of the classifiers must agree on the target class; the third model could have 0 confidence. However, before we spend any extra effort developing a stronger loss function we test this simple loss function.

Surprisingly, we find that it is effective. By running 100 iterations of PGD on this loss function, we are able to reduce the accuracy of the defense to $1.5\%$ at $\varepsilon = 0.031$ (the original evaluation reported accuracy of $13.5\%$ at $\varepsilon = 0.1$ with 40 iterations of PGD).

**Lessons Learned.**

1. The attack themes we distill in Section 4 are not hard-and-fast rules: it is possible to evade defenses while also (knowingly) disregarding them.
2. Ensembles of weak defenses still appear weak [HWC$^+$17].

## J   Temporal Dependency

This defense [YLCS19] detects adversarial examples for automatic speech recognition.

### J.1   How The Defense Works

Given an audio sample $x \in [-1, 1]^t$ represented as the raw audio waveform, an automatic speech recognition system generates a transcription of the input as $y \in \mathcal{Y}^n$ where $\mathcal{Y}$ is the output space of tokens—in this paper, a character "a" through "z" or the whitespace token.

In order to detect if a given input is adversarial, the detection procedure checks if the classification of the first half of the audio waveform is similar to the first half of the classification of the complete audio waveform. Formally, let $s_{1..k}$ denote the prefix of length $k$ for a sequence $s$, and let len$(s)$ return the length of a sequence. The defense begins by setting $x_{\text{pre}} = x_{1..k}$ (the method for selecting $k$ will be discussed later). Then, compute the predicted classification $y = f(x)$ of the full audio waveform $x$ and $y_{\text{pre}} = f(x_{\text{pre}})$ as the prediction of the length-$k$ prefix of $x$. From here, let $l = \text{len}(y_{\text{pre}})$ and compute $\phi = \text{sim}(y_{1..l}, y_{\text{pre}})$ where sim computes the similarity between $y_{1..l}$ and $y_{\text{pre}}$ according to some metric. If $\phi$ is large then the audio is determined to be adversarial; otherwise if $\phi$ is small then the audio is said to be benign.

In order to compute the similarity, the paper proposes various metrics sim; because they all perform roughly the same, we choose the simplest to evaluate: the character error rate. That is, we compute the Levenshtein (edit) distance between the two strings. From source code we obtained from the authors we found that the only detail in the implementation is that the edit distance omits whitespace characters before computing the distance.

It only remains to discuss how to compute $k$. The majority of the paper considers the case where $k = t/2$, so $x_{\text{pre}}$ is the first half of the audio sample. The authors additionally study the cases were $k$ is selected either from some small set of values, or where $k$ is sampled uniformly at random from $0.2$ to $0.8$. In order to attack the most difficult version of the defense we fool the full randomized version of the detector.

**Why we selected this defense.**   This is the only defense that we have seen at ICLR, ICML or NeurIPS that is not focused on image classification; we therefore consider it an interesting case study.

## J.2 Initial Hypotheses and Experiments

The paper performs an extensive adaptive attack evaluation, trying three different attack techniques. One of these attacks looked almost exactly like what we believe would be the correct attack: select a target sequence $t$, and let $t_{\text{pre}}$ denote the prefix of $t$. Then, set the loss function to minimize the loss on both the transcription of the prefixed input $f(x_{\text{pre}})$ and on the full input $f(x)$, where the loss on the prefixed input is determined by the loss between the classification of $f(x_{\text{pre}})$ and the prefix of the target sequence $t$. We noticed one potential failure mode. First, a single hyper-parameter controls the relative importance of both loss terms: the minimized loss function is:

$$\|\delta\|_2 + \lambda \cdot \big(L(f(x'), t) + L(f(x'_{\text{pre}}), t_{\text{pre}})\big) \,,$$

instead of what we would have expected:

$$\|\delta\|_2 + \lambda_1 \cdot L(f(x'), t) + \lambda_2 \cdot L(f(x'_{\text{pre}}), t_{\text{pre}}).$$

We were confident that there was *some* error in the evaluation because the success rate of the attack is low for an *unbounded* attack: even when never placing a bound on the total distortion the adversarial is allowed to make, the attack does not succeed $100\%$ of the time at fooling the detector. Such an unbounded attacker should always *eventually* succeed, if only by adding so much noise that the original audio is completely unrecognizable.

Our guess as to why this may happen comes from our experience with the original attack codebase, which is *not* a simple attack and can thus be hard to debug. Because the objective of the attack is to generate adversarial examples of minimal distortion, the default attack is implemented as follows. First, choose an arbitrary distortion bound $\tau$ that is sufficiently large. Then, perform PGD [MMS$^+$17] on this distortion-bounded input space. Once the attack is successful at fooling the classifier, the distortion bound is reduced by 10% and the attack repeats. There are a number of details here which could have complicated the attack process, however because the authors do not release source code for the attack we are unable to confirm our hypotheses.

## J.3 Final Robustness Evaluation

We implemented the attack as described in the paper, beginning with just one hyper-parameter $\lambda$—we only wanted to introduce a new hyper-parameter if it turned out to be necessary. Specifically, we perform gradient descent on the loss function $L(x') = L(f(x'), t) + L(f(x'_{\text{pre}}), t_{\text{pre}})$ where $t_{\text{pre}}$ is computed as the first $l$ characters of the *true* prediction $t$ where $l = \text{len}(f(x'_{\text{pre}}))$. Importantly, this is *not* just the first $l$ characters of $f(x')$: this value might not be the correct output at any point in time. If we try to send $f(x'_{\text{pre}})$ towards the current prediction $f(x')$ the two loss functions will be in conflict with each other. However, because we know that $f(x')$ will eventually equal $t$, we send $f(x'_{\text{pre}})$ to the prefix of $t$. Once both the prediction of the model reaches our desired target phrase and the detector is fooled completely (i.e., $f(x'_{\text{pre}}) = y_{\text{pre}}$ exactly) we reduce the distortion bound and continue our search.

In order to ensure that our adversarial examples remain effective under random choices of the prefix length $k$ we run our attack by, in parallel, ensuring that the attack is successful with $k = .25$, $k = .5$ and $k = .75$ all simultaneously. We hope from here that this implies robustness to values in between (and empirically we find it does).

There are two important differences between this evaluation and all other evaluations in this paper. First, we perform a targeted attack here: untargeted attacks are uninteresting on audio classification because they typically just induce misspellings. Second, because we run an unbounded attack, instead of reporting the success rate at a given distortion bound, we report the median distance to generate a successful perturbation. Success rates for unbounded attacks should always be $100\%$ and are not valid points of comparison: given sufficient distortion, *eventually* we will succeed. Indeed, we succeed in generating adversarial examples successfully that reach the target class $100\%$ of the time, and that fool the detector $100\%$ of the time.

On a baseline classifier, we require a median $\ell_\infty$ distortion of 41 (out of a 16-bit integer range of 65536). For the classifier defended with this defense, we require a distortion of 46, a slight increase but not significantly higher than baseline defenses (e.g., bit quantization).

**Lessons Learned.**

1. Subtle differences in attack implementations can distinguish between effective and ineffective attacks.
2. Defenses that do not rely on principled inclusion of randomness rarely benefit from its addition.

# K    Mixup Inference

This paper [PXZ20] employs a stochastic interpolation procedure during inference in order to mitigate the effect of adversarial perturbations.

## K.1    How The Defense Works

The defense works as follows. For each input $x$ (e.g. a clean or perturbed image) we compute $K$ interpolations with samples $s_k$,
$$\tilde{x}_k = \lambda x + (1 - \lambda)s_k.$$
where $\lambda$ is a fixed hyper-parameter ($\lambda = 0.6$ in all experiments) and $s_k$ is sampled randomly from a predefined set of images $\mathcal{S}$. We then average the logit responses $z(\cdot)$ of the undefended model over all $K$ interpolations, i.e. the final response $\hat{z}(\cdot)$ of the defended model is

$$\hat{z}(x) = \frac{1}{k}\sum_{k=1}^{K} z(\tilde{x}_k) = \frac{1}{k}\sum_{k=1}^{K} z(\lambda x + (1 - \lambda)s_k).$$

The paper develops two instantiations of the defenses: OL (other label) and PL (predicted label). The two types only differ in how $s_k$ is sampled from $\mathcal{S}$. For OL we sample $s_k$ uniformly from all images for which the predicted label is *different* from $x$. For PL we sample $s_k$ uniformly from all images for which the predicted label is *same* as for $x$. The defense is applied with $K = 1$ during training and with $K = 15$ (OL) or $K = 5$ (PL) during testing.

The motivation for this defense is two-fold: for one, the effect of a perturbation $\delta$ is reduced by $\lambda$ due to the interpolation. Second, the manuscript hypothesizes that a perturbation $\delta$ tends to have an effect on the model only close to a given input $x$ and that the large shifts towards other samples breaks the effect of $\delta$ on the model.

**Why we selected this defense.**    Defenses that employ stochastic elements are notoriously difficult to evaluate for two reasons. First, most attacks implicitly assume the targeted model to be deterministic. Breaking this assumption is a common cause for attack failure. Second, the quality of an adversarial perturbation now has to be measured as the average success rate, i.e. how often it can fool the model, which makes it difficult to select good adversarial perturbations (e.g. across repetitions and attacks). The Mixup defense is a particularly interesting instantiation of a stochastic defense because of its non-local mixing mechanism that interpolates between distant images.

## K.2    Initial Hypotheses and Experiments

A large portion of the paper's main body is dedicated to an oblivious threat model in which the attacker is assumed to be unaware of the defense. This is not an interesting scenario: defending against a naïve attacker is not hard. Evaluating such a scenario is only interesting as a sanity check, but we recommend against presenting these results in the main body.

There is a brief discussion of adaptive attacks. For an adversarially trained model, the results are very similar to the oblivious attack, which raises a red flag: an adaptive attack should almost always perform substantially better than an oblivious attack that ignores the defense. This behavior suggests that the adaptive attack is not well suited for this defense. The exact algorithm of the adaptive attack is left unclear: the paper describes the adaptive attack as an average across multiple adversarial examples generated by PGD against the same sample. In the paper's appendix, however, the adaptive attack is described as a single PGD attack in which the gradient is estimated across multiple linear mixtures. The source code reveals that the first method was used, and that is problematic. Averaging across adversarial perturbations that have each been produced by an ineffective attack is unlikely to

lead to much better adversarial examples. A stronger approach would be to average the gradient in each step of the attack.

Taken together, these observations suggest that a correctly adapted attack algorithm that takes the defense mechanism into account might be able to break the defense.

### K.3 Final Robustness Evaluation

We perform our evaluation on CIFAR-10 using the code released by the authors. The threat model we use is $\ell_\infty$ attacks bounded by $\epsilon = 8/255$. We evaluate the paper's strongest base model (Interpolated adversarial training) which reaches 42.5% accuracy on untargeted attacks without the defense, compared to 63.4% (oblivious attack) or roughly 57% (adaptive attack) for the defended model (MI-OL). We only evaluate the OL defense as suggested by the authors (private communication).

We include the mixing step into our attack. During the attack we backpropagate through the complete mixing mechanism using the same number of samples as used during evaluation. We applied PGD with 50 steps and a step-size of 1/255. During the attack and during evaluation of the perturbations we choose the mixing images randomly in every step. This procedure was sufficient to reduce the accuracy of the model to 43.9%, very close to the accuracy of the base model without the Mixup defense. It is likely that one can further reduce accuracy through additional repetitions or steps, but due to the computational overhead of the Mixup mechanism, we did not perform additional experiments.

**Lessons Learned.**

1. The majority of the evaluation should focus on adaptive attacks, and not on attacks that are oblivious to the defense mechanism. Experiments with oblivious attacks are helpful sanity check and we recommend that authors include them in an appendix.

2. For stochastic defenses, rather than averaging over the results of $n$ failed gradient-based attacks, it is better to stabilize the gradients of the model by averaging over $n$ queries. Only a stable gradient will allow an iterative gradient-based attack to succeed.

## L   ME-Net

This paper [YZKX19] proposes a pre-processing step that randomly discards a large fraction of pixels in an image, and then uses matrix-estimation techniques to reconstruct the image. The defense trains a model on such pre-processed inputs, with the aim of learning representations that are less sensitive to small input variations.

### L.1   How The Defense Works

Given an image $x$ represented as a matrix $M$, the defense first drops each entry in $M$ independently with some probability $p$ to obtain a noisy matrix $N$. It then reconstructs a matrix $\hat{M}$ from $N$ that should be close to $M$ in expectation. Various matrix-estimation techniques are proposed. The ones we focus on here reconstruct $M$ using universal singular value thresholding (USVT) [C$^+$15] or nuclear-norm minimization [CR09].

To train a model, $n$ random noisy matrices are generated for each training input $x$, and the respective matrix reconstructions $\hat{M}^{(1)}, \dots, \hat{M}^{(n)}$ are added to the training set. A standard classifier is then trained on this expanded dataset. At inference, a single random noisy matrix is generated. The classifier processes the reconstruction of that matrix.

To combine the defense with adversarial training, the model is trained on adversarial examples from a PGD attack [MMS$^+$17] that uses the BPDA technique [ACW18] to backpropagate through the matrix estimation step. Here, instead of pre-generating $n$ matrix reconstructions for each training input, a noisy matrix and corresponding matrix reconstruction is computed for every step of PGD.

**Why we selected this defense.**   There have been a number of defenses proposed with a similar randomized pre-processing step [PMG$^+$18, GRCvdM18] which have been broken by adaptive attacks [ACW18, AC18]. Yet, this paper does perform a thorough adaptive evaluation and concludes

that the defense is sound. We thus viewed this defense as an interesting case-study: if we succeed in finding an attack on it, why did the adaptive attacks on similar prior defenses fail?

## L.2 Initial Hypotheses and Experiments

Our first impression from this defense was simply that the proposed idea is too similar to previously broken defenses. At the same time, the paper does perform a fairly thorough evaluation of the defense, against a combination of black-box, white-box and adaptive attacks. Taking a closer look at this evaluation, we notice a few areas of concern.

Since the defense's pre-processing is non-differentiable, the main white-box attack considered in the paper uses PGD with the BPDA technique of [ACW18]. To approximate gradients, the pre-processing is applied in the forward pass, but replaced by the identity function in the backward pass. The rationale for applying BPDA is that the pre-processing step (i.e., randomly dropping pixels and applying matrix estimation) computes a function that is close to the identity. While this use of BPDA appears appropriate, the paper does not take the most appropriate steps to deal with the defense's randomness. The defense is evaluated on attacks with multiple random restarts, but this is typically insufficient if the gradient computations at each step of the attack do not account for the defense's stochasticity. In the same vein, the paper reports success in defending against the score-based attack of [UOOK18] and the decision-based attack of [BRB18], both of which fare poorly against randomized defenses.

The paper considers two adaptive attacks. The first finds an adversarial perturbation for the reconstructed image, and then applies it to the original image. The second applies BPDA and then projects the gradient onto a low-rank subspace so that the attack focuses on global image structures. Both these attacks are found to perform significantly worse than PGD with BPDA, which demonstrates that they are not the most appropriate adaptive attacks.

We formulate the following hypotheses as to how this defense could be broken:

1. The attack should average over the randomness of the defense, using EOT (see Appendix A).

2. The paper seems to apply BPDA in a coarse fashion. As the initial randomized masking step is differentiable, a stronger attack might use BPDA to only approximate gradients of the matrix-estimation computation, rather than of the full pre-processing.

3. Score-based attacks [IEAL18, CZS$^+$17] might provide better gradients than BPDA, if combined with EOT.

We evaluate the defense on CIFAR-10, using code and models shared by the authors. We use the threat model from the paper, $\ell_\infty$ attacks with $\epsilon = 8/255$. We focus our evaluation on the USVT approach, for which we could obtain a standard model as well as an adversarially trained model from the authors. We call these models USVT and USVT$_{\text{ADV}}$. After completing our evaluation, we further trained a standard model using nuclear-norm minimization and tested our best attack on that model as well.[14]

We first reproduce the paper's white-box attack that uses BPDA to approximate the entire pre-processing step with the identity function. The approximated gradient is used in a PGD attack with 200 steps. This attack reduces the models' accuracy to 35% for USVT and 50% for USVT$_{\text{ADV}}$, which is consistent with results reported in the paper.

We then attempted an attack that combines BPDA with EOT. We approximate the gradient 40 times with different random masks, and use the average gradient in a PGD attack. This reduced the models' accuracies to 15% for USVT and 32% for USVT$_{\text{ADV}}$. Note that the accuracy for USVT$_{\text{ADV}}$ is lower than that obtained with adversarial training alone (the same ResNet-18 model architecture achieves over 40% robust accuracy with adversarial training). The reason for this is probably because the attack used during training (PGD+BPDA) is not strong enough to approximate worst-case adversarial examples for this defense.

## L.3 Final Robustness Evaluation

We were curious as to why the BPDA+EOT attack did not have an even higher success rate for the standard model trained with USVT matrix estimation.

First, we found that the defense's randomness causes significant fluctuations in the model's outputs: when classifying inputs multiple times in a row, we find that there's a non-negligible chance of the model misclassifying an input due to a "bad" choice of random mask. This brittleness of the defense may also make it harder to produce robust adversarial examples.

We further investigated whether the assumption underlying the use of BPDA was valid, namely that the pre-processing step approximates the identity function. Surprisingly, we found that this is not necessarily the case for the USVT approach. For CIFAR-10 images, USVT fails to reconstruct an image close to the original. In fact, we find that the output of the pre-processing step is roughly identical to the randomly masked image. We thus tried to apply BPDA only to the USVT procedure, and backpropagate through the random mask. However, this attack did not improve upon the $15\%$ accuracy we reached when applying BPDA to the full pre-processing step (in fact, applying BPDA solely to USVT resulted in a slightly worse attack).

Taking a step back (and while trying to find bugs in our own attack implementation), we attempted to attack solely the trained classifier (a ResNet-18), without the pre-processing stage. Surprisingly, we found the base classifier to be moderately robust: For $\ell_\infty$-noise bounded by $\epsilon = 8/255$, we could not get its accuracy below $10\%$. We further checked the model's gradients, and found them to be highly interpretable, a phenomenon previously demonstrated for adversarially trained models [TSE+19]. The non-trivial robustness of this base classifier is quite remarkable, as it was trained in a standard fashion on randomly masked and reconstructed images. This gives credence to the authors' intuition that training on randomly masked and reconstructed images can encourage a model to learn somewhat more robust representations. Altogether, this suggests that an attack on the full defense is unlikely to bring the model's accuracy much further down than our best attack.

Finally, we evaluated a model trained with nuclear-norm minimization. Due to the high computational cost of that defense (the nuclear-norm minimization is CPU-bound and takes about one second per input), we refrained from performing any additional adaptive experiments, and simply used the best attack we found for the USVT model (i.e., BPDA+EOT). This attack reduced the accuracy of the nuclear-norm defense to $13\%$, a similar result as for the defense instantiated with USVT.

**Lessons Learned.**

1. Attacks must account for all of the randomness of a defense.
2. For defenses that train a base classifier with additional components, an ablation study can reveal where the increased robustness comes from.

# M  Asymmetrical Adversarial Training

This paper [YKR20] proposes to use adversarially-trained models to detect adversarial examples. The idea is a natural extension to the work of [MMS+17].

## M.1  How The Defense Works

For a $K$-class problem, the defense uses $K$ detector models $h_1, \ldots, h_K$. Given an input $x$, the $i$-th detector outputs a logit score $h_i(x) \in \mathbb{R}$ for class $i$. Let $\sigma(\cdot)$ denote the sigmoid function. The detectors are adversarially trained so that for each training input $(x, y)$, we maximize $\sigma(h_y(x))$ and minimize the terms $\max_{||\delta|| \le \epsilon} \sigma(h_i(x + \delta))$, for all $i \neq y$. The first term ensures that the detector of the correct class $y$ recognizes $x$ as benign. The other terms encourage the detectors of classes $i \neq y$ to reject perturbed versions of $x$.

The paper proposes two classification frameworks:

1. The *integrated classifier* uses a standard "base" classifier $f(x)$ to output a prediction $f(x) = y$. The input is then fed to the detector for that class, $h_y$, and rejected if the logit $h_y(x)$ is below some threshold $\tau$.

2. The *generative classifier* computes a logit vector from the scores of all detectors. The classifier's prediction is $f(x) = \arg\max_i h_i(x)$ and low confidence inputs are rejected (i.e., if $\max h_i(x) < \tau$ for some threshold $\tau$).

**Why we selected this defense.** This defense is interesting in that the authors seemed to have done everything right. The proposed adaptive attack (see below) is well motivated and introduces no unnecessary complexity. We initially believed that this defense would resist our own adaptive attacks.

## M.2 Initial Hypotheses and Experiments

Compared to many other defenses, this ensemble of adversarially-trained detectors is fairly simple (a positive trait in our opinion). The classifier is built from $K$ standard models, none of which should (a priori) show any signs of gradient masking or other optimization difficulties. The defense's evaluation is also quite appropriate. The detectors are first evaluated in isolation using standard attacks, and the authors further propose an adaptive attack with a loss function tailored to the full defense. Finally, the authors show that their robust detectors are *invertible*. That is, an unbounded attack that maximizes the score of a detector results in semantically meaningful images from that class—mirroring similar results for adversarially trained classifiers [STT+19]. Overall, this gives good credence that this defense is indeed robust.

A closer look at the defense's evaluation does however reveal one issue: a sub-optimal choice of loss function in the considered adaptive attack. Let $z(x)_i \in \mathbb{R}$ denote the $i$-th logit of the base classifier. Given an input $(x, y)$, the attack on the integrated classifier finds an adversarial example $x'$ that maximizes:

$$L(x', y) = \begin{cases} \max\limits_{i \neq y} z(x')_i - z(x')_y & \text{if } f(x') = y \\ \max\limits_{i \neq y} h_i(x') & \text{otherwise} \end{cases} .$$

In words, the attack first finds an $x'$ that is misclassified by the base classifier, using a standard loss function proposed in [CW17b]. Then, once the base classifier is fooled, the attack further maximizes the scores of all detectors other than that of the true class $y$, to ensure that the example is not rejected.

The issue with this loss function is that the second term, $\max_{i \neq y} h_i(x')$, does not capture the true attack objective. Indeed, to bypass the defense, we only need to fool the detector of the class predicted by the base classifier, i.e., $y' = f(x')$. Maximizing the scores of all detectors is thus wasteful of the attack's limited perturbation budget.

The adaptive attack on the generative classifier has a similar issue. Here, the proposed loss function is simply:

$$L(x', y) = \max\limits_{i \neq y} h_i(x') .$$

That is, the attack maximizes the score of the most confident detector other than the one of the true class. The issue is that even with an unbounded perturbation, maximizing this loss may fail to produce a successful attack. For example, one way to maximize this loss function is to increase each detectors' score by the same value, thereby leaving the classifier's prediction unchanged.

## M.3 Final Robustness Evaluation

We evaluate pre-trained CIFAR-10 detectors released by the authors. We report results for $\ell_\infty$ attacks with $\epsilon = 8/255$.

We first describe our attack on the integrated classifier. A natural idea for improving the loss function from the paper would be to first fool the base classifier into outputting some class $f(x') \neq y$, and then to maximize the score of the detector for class $f(x')$:

$$L(x', y) = \begin{cases} \max\limits_{i \neq y} z(x')_i - z(x')_y & \text{if } f(x') = y \\ h_{f(x')}(x') & \text{otherwise} \end{cases} .$$

However, we tend to dislike loss functions of this type as work is still wasted if the classifier's prediction $f(x')$ ever switches from some class $i \neq y$ to another class $j \neq y$. Instead, we formulate a

*targeted* attack objective:

$$L_t(x') = z(x')_t - \max_{i \neq t} z(x')_i$$

$$L(x', t) = \begin{cases} L_t(x') & \text{if } L_t(x') < \kappa \\ h_t(x') & \text{otherwise} \end{cases} \quad .$$

That is, we first fool the base classifier into outputting the target class $t$, and then maximize the score of the detector for the target class. The small constant $\kappa > 0$ ensures that $f$'s targeted misclassification is not overly brittle. We stop the optimization on the base classifier $f$ once a targeted misclassification is achieved since the confidence of $f$ is not considered by the defense.

We then perform a *multi-targeted* attack [GUQ+19, Car19]: for each target class $t \neq y$, we maximize $L(x', t)$ using 100 PGD steps, and we retain the best adversarial example across all $K - 1$ attacks.

This attack is substantially more effective than the one in the paper's evaluation. At a False-Positive-Rate of 5%, we reduce the detector's robust accuracy from 30% to 11%.

For the generative classifier, we also used a multi-targeted attack as it is typically very effective [GUQ+19]. Our objective maximizes the score of the targeted detector, while minimizing the highest detector score, until the targeted detector's score is highest. At that point, we further maximize the target detector's score to ensure a confident prediction:

$$L_t(x') = h_t(x') - \max_{i \neq t} h_i(x')$$

$$L(x', t) = \begin{cases} L_t(x') & \text{if } L_t(x') < \kappa \\ h_t(x') & \text{otherwise} \end{cases} \quad .$$

At a False-Positive-Rate of 5%, the multi-targeted attack (with 100 steps of PGD per target) reduces the generative classifier's robust accuracy from 55% to 37%, lower than that of full PGD adversarial training.

This suggests that the attacks used at training time might not be optimal either. Recall that the defense is trained so as to maximize $\sigma(h_y(x))$ and minimize $\max_{||\delta|| \leq \epsilon} \sigma(h_i(x + \delta))$ for $i \neq y$. Intuitively, while this ensures that the detectors for class $i \neq y$ have low confidence on adversarial examples, it does not ensure that the detector for the true class has high confidence on adversarial examples. Adversarially training *all* detectors for each input might lead to a more robust defense.

Given that our attacks did not fully reduce the accuracy of either model to 0%, one may wonder why we are confident that our evaluation is sufficiently thorough that we have accurately evaluated the defense's robustness to within a small margin. As this defense is not certified, our evaluation will ultimately always be empirical. We believe it unlikely we can significantly reduce the defense's accuracy further due to the following:

- The defense's simplicity, which does not suggest any gradient masking issues.
- The defense's strong interpretability—as demonstrated by the authors—which had been previously observed in other robust models.
- Our attacks' simplicity, which reduces the risk of implementation errors or of wasting perturbation budget on unnecessary objectives.
- The defenses' resilience under standard checks [CAP+19]: increasing the number of PGD iterations (to 200) and random restarts (to 10), only reduce the defense's accuracy by an extra 1%.

**Lessons Learned.**

1. A good loss function has the property that increasing the loss always increases the attack success rate. Loss functions without this property can "waste" the perturbation budget on orthogonal objectives.
2. It is sometimes easier to write down a good targeted loss function than an untargeted one. A "multi-targeted" attack that targets each class in turn will be more effective than an untargeted attack in such a case.

# N   Turning a Weakness into a Strength

This defense [HYG$^+$19] is an adversarial example detector. While the defense is conceptually simple, the original evaluation methodology is complex and constructing a single unified loss function that encodes the attacker's objective is challenging.

## N.1   How The Defense Works

Note that in this section, we sometimes use $f(x)$ to refer to either the classifier's output class, or the full vector of class probabilities. The choice should be clear from context. We denote by $f(x)_i$ the probability that the classifier assigns to class $i$. To determine if a given input $x$ is adversarial, the defense sets thresholds $\tau_1$ and $\tau_2$ and checks if either of the following are true:

1. $\sum_i \|f(x+\delta)_i - f(x)_i\|_1 > \tau_1$ for $\delta \sim \mathcal{N}(0, I\sigma^2)$; or,

2. `#Steps`$(f, x) > \tau_2$ where `#Steps` returns the number of steps required to generate an adversarial example $x'$ such that either (a) $f(x) \neq y$ or (b) $f(x) = t$.

The first term is simple: the models predictions should not change substantially when the input is slightly perturbed by Gaussian noise. An idea related to this term alone has previously been suggested as a defense [CG17] and was subsequently broken [HLS18].

The second loss term is more complicated: given an input $x$ we compute the number of steps necessary to turn $x$ into an adversarial example $x'$. Intuitively, this check enforces that $x$ is *not* high confidence: if it is too far from the decision boundary, then this test will fail. This would have prevented He *et al.*'s attack.

**Why we selected this defense.**   This defense is conceptually simple, but by construction it is hard to define a straightforward differentiable loss function that can be minimized to generate adversarial examples against it. This defense relies on ideas similar to the defense in Appendix C.

## N.2   Initial Hypotheses and Experiments

The paper contains an adaptive attack by developing a new loss function that is designed to evade the defense. This loss function is defined as a combination of four independent loss terms $L = \lambda L_1 + L_2 + L_3 + L_4$.

The first term $L_1$ is a standard cross-entropy loss to fool the classifier, $L_1 = L_{CE}(f(x), y_{\text{true}})$. The second loss term encourages the classifier to be confident in its prediction by directly optimizing the first detection term $L_2 = \mathbb{E}_{\delta \sim \mathcal{N}(0, I\sigma^2)}[\|f(x) - f(x'+\delta)\|_1]$ in order to defeat the first detector. The third loss term encourages the classifier to have a high loss after taking a single gradient descent step of size $\alpha$ towards any incorrect class $y'$ $L_3 = -\mathbb{E}_{y' \neq y_t} L_{CE}(f(x - \alpha \nabla_x L_{CE}(f(x), y')), y_t)$ and the final loss term encourages the same but for the target class $y_t$. $L_4 = -L_{CE}(f(x + \alpha \nabla_x L_{CE}(f(x), y_t)), y_t)$ Loss functions $L_3$ and $L_4$ are non-differentiable, and so the paper applies BPDA [ACW18].

This combined loss function is exceptionally complicated, and it is unlikely that optimizing it would be able to minimize all four terms simultaneously. The lack of additional weighting terms $\lambda$ on each hyper-parameter made us suspect that when generating an adversarial examples, $L_1$ would dominate and the remaining loss terms would not actually change the final resulting adversarial example. Combining just two terms together can often be difficult [CW17a] and requires adaptively tuning the hyper-parameter $\lambda$; combining four introduces even more complexity.

The paper applies PGD using the standard cross-entropy loss [MMS$^+$17] and the logit margin loss [CW17b]. Typically, the difference between these two losses is small: a few percentage points or less. However this paper finds a large difference in many settings. For example, at a distortion bound of $\epsilon = 0.1$ the PGD on the cross entropy loss succeeds just $8.5\%$ of the time. In comparison, the margin loss succeeds over $32\%$ of the time. This large of a gap is atypical; we suspected the root cause to be a non-standard experimental methodology or attack implementation.

As a result, we inspected the open-source implementation provided by the authors. The paper implements its own attack algorithms instead of calling a primitive from some reference attack library

such as Foolbox [RBB17] or CleverHans [PFC+16]. We find that the implemented attack does not take the sign of the gradient when implementing $\ell_\infty$ PGD [MMS+17]. The typical definition of $\ell_\infty$ PGD is to define $\text{normalize}(x) = \text{sign}(x)$. The implementation of PGD contained in this paper does not include the sign operator and does *not* normalize the gradient. Importantly, the paper makes no definition of PGD and as such does not explicitly state if their PGD method includes this normalization step.[15]

Thus, this explains the artifact we observed where the logit margin loss is superior to cross-entropy loss. The magnitude of the gradient of the cross entropy loss can vary by several orders of magnitude. When the classifier is confident in its prediction, as is typically the case for poorly-calibrated models, $\|\nabla L_{CE}\|$ is often exceptionally small: $10^{-5}$ or lower. In contrast, when the classifier is not confident and the top two predictions are almost equally likely, $\|\nabla L_{CE}\|$ can be as large as 1. The gradient of the margin loss is much better behaved: it tends to not change by more than an order of magnitude.

More concerning than the fact that the implementation of PGD used to attack the network does not contain the sign operator, the implementation of PGD used in phase (2) *of the defense* (that computes #Steps) omits the sign operator. Because the defense uses the number of steps of gradient descent as a proxy for the $\ell_p$ distance to the decision boundary itself (instead of using the distance to the decision boundary directly), this implementation difference is extremely important. The defense would probably have been more effective if it had used the standard attack implementation with the sign on the gradient.

### N.3 Final Robustness Evaluation

The core assumption the defense relies on is that there do not exist inputs that (a) have high confidence on random noise, but (b) are still close to the decision boundary when running an adversarial attack. One option would be to mount a feature adversary attack as we did against a similar defense in Appendix C. However, in this section we aim to mount a new type of adaptive attack. We set out to directly construct an input with properties (a) and (b) above, by initially generating an adversarial example with high confidence and then slowly moving it towards the decision boundary.

For an input example $x$ we initially generate a high-confidence adversarial example $x'$ by running 100 steps of PGD (*with* the sign on the gradient). Then, we let $x_\alpha = \alpha \cdot x' + (1 - \alpha) \cdot x$ linearly interpolate between the original example and the adversarial example, such that $x_0 = x$ and $x_1 = x'$. We find that often, there exists a choice $x_{\alpha^*}$ that it is still high confidence but also is sufficiently close to the decision boundary that the defense will not reject it.

Our first attack simply samples 1,000 values of $\alpha$ between 0 and 1 and checks if the defense's checks (1) and (2) are both satisfied for any of these examples. There are two problems with this approach. First, it is computationally expensive: the detection method has an inline loop that runs PGD and therefore to generate a single adversarial example requires several minutes. Worse, we find that it sometimes fails when a more fine-grained search would have succeeded, because the necessary value of $\alpha$ to fool the defense is very small, i.e., $\alpha < 10^{-4}$. Increasing the number of steps is one possible solution—which we used to diagnose the failure here—but is even slower.

To improve the computational complexity, we instead perform a binary search to find a value of $\alpha$ so that both defense properties (1) and (2) hold. We initialize $\alpha_{\text{low}} = 0$ and $\alpha_{\text{high}} = 1$. We define $\alpha_{\text{mid}} = (\alpha_{\text{low}} + \alpha_{\text{high}})/2$ and evaluate $x_{\alpha_{\text{mid}}}$. If property (1) holds but property (2) does not, then we know that the current input is robust to noise but too far away from the decision boundary, and thus we would like to move it closer to the boundary. So we set $\alpha_{\text{low}} = \alpha_{\text{mid}}$ and recurse. Conversely, if property (1) does not hold but property (2) does hold, then we are too close to the decision boundary and need to back away. We set $\alpha_{\text{high}} = \alpha_{\text{mid}}$ and recurse. If both properties hold, we have a successful adversarial example. If neither hold, we randomly generate a new adversarial example $x'$ and repeat the binary search procedure for this example.

The most important reason why this attack is effective is that it is simple to analyze. It can be difficult to diagnose why gradient descent fails, and so designing attacks that are as simple as possible makes it easy to diagnose why they are not working effectively. When simple techniques do not work as expected, it is easy to learn the correct lesson for why they failed. In contrast, when complex

techniques do not work as expected, one has to consider multiple hypotheses and evaluate each one to understand the true reason for failure.

This attack is successful at reducing the accuracy of the classifier on CIFAR-10 to $0\%$ at a $0\%$ detection rate, and similarly bring the ImageNet classifier to $< 1\%$ accuracy at a $0\%$ detection rate, both in the threat model originally considered by the paper.

**Lessons Learned.**

1. Combining multiple terms together leads to difficult-to-optimize loss functions that may not behave as desired.
2. BPDA should not be treated as a general-purpose method for minimizing through arbitrarily non-differentiable functions. Rather, loss functions must be designed to work with BPDA and must already be *almost differentiable*.

## Footnotes

[7]Note that an adaptive attacker could just remove such a non-differentiable layer and then succeed. While such a defense may sound implausible, Appendix G evaluates exactly such a defense that unintentionally post-processes with the "identity" function $g(y) = \log(\exp(y))$ which, while mathematically a no-op, causes numerically unstable gradients. Removing this post-processing function breaks the defense.

[8]The values $\bar{\Delta}_{y,i}(x)$ are further normalized using means and standard deviations computed on the training set. This step is unimportant to understand the defense and our attacks.

[9]We break the defense in Appendix N with a single-purpose attack that is not as generalizable as the one described in this section, but which demonstrates the wide space of adaptive attacks.

[10]A description of each of these attack techniques is in Appendix A.

[11]Our notation differs slightly from that in the paper, which denotes the latent variable by $z$, because we use $z$ for the logit vector.

[12]Deep generative models tend to be much more complex than standard discriminative models, so the complexity of this defense is not unwarranted. But this does make a strong adaptive evaluation challenging.

[13]The paper generates the matrix $A$ using some procedure that is unimportant for understanding the defense.

[14]Adversarial training with the nuclear-norm approach is extremely expensive (on the order of 10 CPU years) and we could not obtain a pre-trained model for this setting. So we refrain from evaluating it.

[15]In personal communication with the authors they state that they intended for PGD to *not* have the sign operator.