[Reviews · NeurIPS 2020]

Review 1

Summary and Contributions: This paper designs adaptive attacks to many adversarial defenses and demonstrates that 13 recent defenses are actually vulnerable to adaptive attacks. This paper provides a thorough analysis on the attack themes and gives suggestions for further similar defenses.

Strengths: A main contribution of this work is to provide guidance on how to evaluate defense methods against adaptive attacks. As there are increasing works on this field, a proper and correct evaluation of their worst-case adversarial robustness is necessary to further advance the field. This works studies 13 defenses, which covers a wide rage of defense techniques.

Weaknesses: One of the main claim of this work is that adaptive attacks cannot be automated, since no single strategy would be sufficient. The authors develop diverse attack strategies to evade different defenses. However, such a finding may not be exactly followed by future works, as the authors of defenses may intentionally or unintentionally neglect strong adaptive attacks. And the "arms-race" between attacks and defenses will still last. So can these failure modes of defenses found in this paper be summarized into several automated attack strategies, such that a principled guidance can be followed in future works. I saw one work on building AutoAttack (Croce and Hein, ICML 2020). For the defenses in this paper, why not using query-based black-box attacks to evade them all? A lot of key technical information is left in the supplementary material. It's a little bit hard to understand how and why the adaptive attacks are proposed and effective for those defenses. But by reading the supplementary material, it's much clearer. This paper mainly evaluate non-adversarial-training-based defenses. A lot of adversarial training methods have been proposed, which may be further evaluated.

Correctness: Yes.

Clarity: Yes.

Relation to Prior Work: Yes

Reproducibility: Yes

Additional Feedback: After rebuttal: Thank the authors for the rebuttal, which addressed most of my concerns. Now the comparisons with AutoAttack is much clearer. It's good to see that the adaptive attacks can outperform AutoAttack in many cases. My suggestion on the improvement on this paper is to summarize the strategies and designs of adaptive attacks, making them easier to follow for future defenses. As the authors listed the attack themes in Table 1, a more detailed suggestion for each of the theme can be presented.


Review 2

Summary and Contributions: The paper analyses a collection of recently published (at top tier conferences) defenses against adversarial machine learning. The result is sobering: None of the 13 defenses survives the presented adapted attacks.

Strengths: In order for adversarial machine learning research to be taken seriously in the security world, defenses that are accepted at top tier machine learning conferences must present a rigorous analysis on any possible attack avenue with respect to their defense. Of course, one may not yet know what attacks are possible, but it is clear that a better job is expected – and this paper shows a way of reasoning and analyzing that can be adapted to bring defense papers to a new solid level.

Weaknesses: It is not clear which adversarial models each attacked defense assumes. To make this work more precise the explanation for each defense should have a description of the assumed adversarial model. Next an attack within that model can be presented. By adding adversarial models, the reader will recognize models that range from weak to strong adversaries. This will also allow the reader to evaluate the strength of an attack. An attack is stronger if it works for weaker adversaries (i.e., adversaries with less capabilities). For example, a pure white box attack assumes a gradient oracle, quite a strong assumption, hence, a strong adversary. A pure black box attack on the other hand only has oracle access to the classifier (and a less pure black box attack may give access to the score vector). Of course black box attacks themselves differ as sometimes an adversary is strong in the sense that the adversary gets access to all the training data (so that an own accurate synthetic model can be learned). In short, for the paper to truly connect to the security community, detailed discussions of adversarial models and strength of attack capabilities needs to be added. Post rebuttal: As a security expert with a long publication track I find this paper to be important in the field of adversarial ML because it shows the low quality of adversarial ML defense publications that are accepted. A proper well thought out security analysis often lacks -- there is a lack of a systematic approach. But to even have this paper be taken seriously by the security community, the authors must explain in rigorous terms what adversarial model is used. Simply saying white box attack is not sufficient: Which oracles does the adversary have access to (an oracle that provides some/all training data, an oracle that provides gradients or score vectors or class labels, etc.)? Notice that in the security community white box is something totally different from what the adversarial ML community has in mind (it is often just allowing an oracle to computing gradients while the so-called black box oracle which provides class labels is very powerful -- in some ways more powerful than the white-box oracle, see the beware the black-box eprint https://arxiv.org/abs/2006.10876 ). So I hope this motivates the authors to do a much better job on being rigorous with respect to adversarial models.

Correctness: Claims and method are correct.

Clarity: Well written.

Relation to Prior Work: Related work is well explained.

Reproducibility: Yes

Additional Feedback:


Review 3

Summary and Contributions: => This paper studies the problem of properly evaluating machine learning models against adversarial example attacks. In particular, the paper advocates for a better evaluation against adaptive attack (i.e. attacks that knows how the defense is being performed). => The authors demonstrate that thirteen defenses recently published at ICLR, ICML and NeurIPS can be broken by adaptive attacks, even though the initial papers claimed to perform evaluations using such (adaptive) attacks. Based on this observation, the paper provides insights and good practices on how to perform an adaptive attack.

Strengths: => The paper is clear and very well written. The claims are clearly identified and simple to understand. => The problem of evaluating empirical defenses against adversarial examples is a burning issue, and this topic is highly relevant to the NeurIPS community. => The empirical evaluation is very clear, and comes with enough details and code to allow reproducibility of the experiments. I also appreciate that the authors took the time to correspond with authors of the initial papers to discuss the defense techniques and to verify the efficacy of the attacks. => The taxonomy provided in section 4 seems novel to me and the defenses illustrate a sufficiently diverse set of strategies for the reader to gain knowledge on how to perform an adaptive attack. => The paper presents some new tricks that are quite interesting e.g. the use of multi-targeted attacks when the objective seem hard to evaluate.

Weaknesses: => To my opinion the main issue of the paper is the significance of the contribution. In fact the problem is not new and a lot papers and technical report already demonstrated that empirical defense often offers no practical robustness against stronger attack than the one the were initially evaluated against [1,2]. Some works also provided general guidelines for better evaluation [3]. Accordingly, I feel like the significance of the paper's contribution is hard to evaluate. I think that the paper would greatly benefit from a proper related work section that would clarify the contributions. => Another weakness of the paper is that defenses are broken by existing techniques. Indeed, at the end of the analysis, most of the defenses are broken either by using EOT, BPDA, or by tuning the parameters of existing attacks such as PGD . Some defenses are broken by using decision based attacks. All this techniques already exist in the litterature [1,2,3,4]; hence the technical part is not novel (see also related work section). [1] Adversarial examples are not easily detected: Bypassing ten detection methods. Nicholas Carlini and David Wagner. ACM Workshop on Artificial Intelligence and Security 2017 [2] Obfuscated gradients give a false sense of security: Circumventing defenses to adversarial examples. Anish Athalye, Nicholas Carlini, and David Wagner. ICML 2018 [3] On Evaluating Adversarial Robustness. Nicholas Carlini, Anish Athalye, Nicolas Papernot, Wieland Brendel, Jonas Rauber, Dimitris Tsipras, Ian Goodfellow, Aleksander Madry, Alexey Kurakin. Living document 2019 [4] Reliable evaluation of adversarial robustness with an ensemble of diverse parameter-free attacks. Francesco Croce, Matthias Hein. ICML 2020

Correctness: => The empirical methodology looks correct to me, as well as the claims. => I just have a small question concerning one of the key messages "adaptive attacks cannot be automated and always require careful and appropriate tuning to a given defense". I do agree with this statement in general. However, I also noticed that a lot of defenses were broken by fine-tuning the parameter of PGD. This means that automated techniques such as the one presented in [4], should break (see related work section for more details) these defenses also. Hence my question: how many of the thirteen papers really require a careful adaptive attack?

Clarity: => The paper is clear and easy to read. However due to the format limitation, most of the subsection in section 5 are very small and leaves most of the really interesting parts to the supplementary materials.

Relation to Prior Work: => The paper is very well documented, and acknowledge a lot of existing works (including [1,2,3]). However I believe that an important related work is missing, namely [4]. This work tackles the issues of improper tuning of hyper-parameters in the attacks, as well as gradient obfuscation/masking. It gives an automated procedure to overcome classical failure mode of PGD attacks such as suboptimal step size and problems of the objective function. The paper also advocates for evaluating adversarial defenses with black box decision based attacks. To prove the efficacy of the framework, the authors benchmark their automated attack method against various existing defenses (overlapping with the present evaluation at least in 4 cases). => To clarify the contributions, I would suggest that the author add a proper related work section that would clarify the contributions w.r.t [2,3] and to thoroughly compare their work with [4].

Reproducibility: Yes

Additional Feedback: => In light of the above comments/questions, I think the paper would benefit from dropping some defenses. Even though I agree that the choice of presenting all results without choosing them might be a good idea for a technical report, describing thirteen papers might be too much w.r.t the format constraints. For the main body of the paper, I think the authors should select only a few interesting defenses, which would allow to present the full analysis. Maybe focus on the points where there is a new idea or a methodological improvement. => Minor comment: there are many referencing issues in the supplementary materials (e.g. lines 620 - 661 -712 -736, and footnotes 7-9-10.) After rebuttal: => The authors answered my concerns on related works both on automated attacks and on previous works advocating for adaptive attacks. Including these discussions would improve the quality of the paper.

[Author Response · NeurIPS 2020]

We thank the reviewers for their helpful comments. Below, we address some of the points made regarding our work's contributions and relation to prior work.

**On automated attacks (Reviewer 1 and 3).** Reviewer 1 and Reviewer 3 argue that "AutoAttack" (Croce & Hein, 2020) is a counter-example to our claim that adaptive attacks are inherently necessary and important to study. However, we believe that the results from that work exactly *support* our claim:

1. **AutoAttack fails to attack many defenses we break completely**. AutoAttack does not fully break the challenging "k-winners take all" defense (19% accuracy), whereas we reduce it to 0% accuracy. Our adaptive attack also outperforms AutoAttack on "ME-Net" and on at least two other defenses we independently evaluated ("Asymmetrical Adversarial Training" and "Are Generative Classifiers More Robust").

2. **AutoAttack only applies to standard feed-forward classifiers.** Many defenses depart from the standard defense template that AutoAttack supports. Of the 13 defenses we study, 5 aim at detecting adversarial examples. For these, formulating an appropriate loss function to optimize is precisely the challenge in developing a strong attack, and it is unclear how this process could be automated by AutoAttack.

   AutoAttack also cannot be directly applied to "Temporal Dependency" (a speech-to-text model) and "Robust Sparse Fourier Transform" (which is aimed at perturbations of small $\ell_0$ norm).

   In fact, a majority of defenses evaluated by AutoAttack use adversarial training, for which there is usually no need for an "adaptive" attack as the inference phase is unchanged. This addresses a question of Reviewer 1, on why we refrained from evaluating such defenses in our paper.

3. **It is easy to build an ineffective defense that AutoAttack fails to break**. First, pick any of the defenses evaluated on AutoAttack for which the query-based attacks (FAB and Square) fail. Then, add a component that masks gradients (e.g., a non-differentiable quantization layer) to defeat PGD. Now AutoAttack fails, even though the defense is easy to circumvent with an adaptive attack.

   (This example also addresses another question of Reviewer 1: since query-based attacks routinely fail, e.g., for randomized defenses, we cannot use them to reliably evaluate all defenses.)

We believe AutoAttack is a strong, *non-adaptive* baseline. However, it is orthogonal to our paper, which argues that static automated attack strategies are not sufficient. The above points illustrate why.

**On adversarial models (Reviewer 2).** We apologize for not clarifying this in the paper. All 13 defenses assume a white-box adversary with full access to the defense parameters. The defenses differ slightly in the perturbation norms and bounds that they consider, and these are mostly incomparable. As a result, all our attacks assume white-box model access and operate with the same perturbation bounds as in the original evaluation. While some of our attacks use black-box optimization techniques, this is solely to side-step gradient-masking. We still view these as white-box attacks.

**On related work & technical novelty (Reviewer 3).** We view the fact that "defenses are broken by existing techniques" as a strength rather than a weakness. Introducing new attack techniques would give the incorrect impression that we currently lack the appropriate technical tools to evaluate defenses properly. But for all of the defenses we studied, a better attack can be built using only tools that are well-known in the literature.

Thus, **the issue with current defense evaluations is methodological rather than technical.** Proposing new attack techniques is not going to fix this. Instead, our paper clearly documents how to make use of existing techniques to build strong adaptive attacks.

This is what differentiates our work from prior work that proposed and argued for adaptive attacks (e.g., Carlini & Wagner 2017, Athalye et al. 2018, Carlini et al. 2019). When those papers were written, defense authors indeed lacked the motivation and tools to conduct strong adaptive evaluations. But of the 13 defenses we study, nearly all claim to perform an adaptive attack evaluation following current best practices. The main contribution of our paper is thus to highlight that there still is a systemic lack of strong adaptive evaluations in this field, and to show how to remedy this.

[Meta-Review · NeurIPS 2020]

his paper designs adaptive attacks to many adversarial defenses and demonstrates that 13 recent defenses are actually vulnerable to adaptive attacks. The original reviews were mixed - e.g., there were concerns on the systematic design/choice of the defenses and the analysis on the experimental findings. However, after several rounds of discussions (the authors also did a good job in rebuttal). the reviewers tend to agree that this paper has its value although being not perfect. So, the final recommendation is to accept the paper to NeurIPS.